# H4K20me1 plays a dual role in transcriptional regulation of regeneration and axis patterning in *Hydra*

Akhila Gungi[1] , Shagnik Saha[2] , Mrinmoy Pal[1] , Sanjeev Galande[1,2]

The evolution of the first body axis in the animal kingdom and its extensive ability to regenerate makes *Hydra*, a Cnidarian, an excellent model system for understanding the underlying epigenetic mechanisms. We identify that monomethyltransferase SETD8 is critical for regeneration in *Hydra* because of its conserved interaction with β-catenin to fine-tune the associated gene regulatory network. Inhibition of SETD8 activity abolishes head and foot regeneration in *Hydra*. Furthermore, we show that H4K20me1, the histone mark imparted by SETD8, colocalizes with the transcriptional activation machinery locally at the β-catenin-bound TCF/LEF-binding sites on the promoters of head-associated genes, marking an epigenetic activation mode. In contrast, genome-wide analysis of the H4K20me1 occupancy revealed a negative correlation with transcriptional activation. We propose that H4K20me1 acts as a general repressive histone mark in Cnidaria and describe its dichotomous role in transcriptional regulation in *Hydra*.

## Introduction

Posttranslational modifications (PTMs) of proteins are major regulators of cellular functions (Deribe et al, 2010), with nuclear histone PTMs being prominent context-specific players in transcriptional regulation. Although the methylation of histone H4 was one of the first histone PTMs discovered (DeLange et al, 1969), the modifiers responsible are recent discoveries. Although several enzymes are known to deposit the di and trimethyl marks on H4K20—SUV4-20H1 and SUV4-20H2 being the predominant ones (Schotta et al, 2008), there is only one known monomethyltransferase—SETD8 (KMT5A or Pr-SET7) (Oda et al, 2009). SETD8 has many functions in cells and acts by disrupting signaling pathways (Ke et al, 2014), regulating transcription factors (Choi et al, 2020), altering the chromatin accessibility around the promoters of genes (Myers et al, 2020), and preventing both oncogene-induced and replicative cellular senescence by suppressing nucleolar and mitochondrial activities (Tanaka et al, 2017). SETD8 is involved directly in the Wnt/β-catenin signaling pathway in mammalian cells, *Drosophila* larvae, and zebrafish (Li et al,

2011). SETD8 is required to activate Wnt target genes by interaction with TCF4 during the development of zebrafish and the wings of *Drosophila* (Li et al, 2011; Yu et al, 2019). The methylation of H4 is highly evolutionarily conserved and exists in three states: mono, di, and trimethylation. Whereas the mono- (H4K20me1) and di-methylated (H4K20me2) H4K20 are involved in DNA replication and DNA damage repair, the trimethylated H4K20 (H4K20me3) is a mark of silenced heterochromatic regions (Schotta et al, 2004; Schotta et al, 2008). The function of H4K20me1 is elusive, with reports of activation (Kapoor-Vazirani & Vertino, 2014; Lv et al, 2016; Shoaib et al, 2021) and repression (Nishioka et al, 2002; Tjalsma et al, 2021) of transcription.

To understand the role of H4K20me1 in transcriptional regulation of developmental processes, including axis patterning and regeneration, *Hydra* serves as an excellent model organism. *Hydra* belongs to the phylum Cnidaria which is the phylum that innovated a body axis during the evolution of multicellular organisms. It also harbours extensive powers of regeneration, which require the recapitulation of the axis patterning gene regulatory networks to be formed de novo. Both embryonic and regenerative de novo axis patterning are regulated by the head organizer Wnt/β-catenin signaling pathway. Although multiple studies have investigated the molecular networks underlying the axis patterning (Bode, 2011; Gufler et al, 2018; Reddy et al, 2019; Vogg et al, 2019; Moneer et al, 2021; Unni et al, 2021), few study the epigenetic regulators of gene expression (Lopez-Quintero et al, 2020; Reddy et al, 2020). We identify that the H4K20me1 writer SETD8 is critical for Wnt-triggered regeneration and that the Wnt pathway regulates both the enzyme and its target modification. We also identify dual modes of transcriptional regulation by H4K20me1, activating transcription at promoters downstream to specific signaling pathways. In addition, at the whole genome level, it excludes activation-associated features, allowing us to describe the presence of a repressive histone mark for the first time in Cnidaria.

## Results

### SETD8 is important for both head and foot regeneration in *Hydra*

In *Hydra*, upon amputation, the injury site triggers a wound-healing response involving reorganization of the epithelial cells in 1 h. After

[1]Laboratory of Chromatin Biology and Epigenetics, Department of Biology, Indian Institute of Science Education and Research, Pune, India   [2]Centre of Excellence in Epigenetics, Department of Life Sciences, Shiv Nadar University, Delhi-NCR, India

Correspondence: sanjeev@iiserpune.ac.in; sanjeev.galande@snu.edu.in

this, morphological changes are only visible from 30 hours postamputation (hpa), and at 30–36 hpa, tentacle buds start emerging, indicating successful differentiation of head structures. The emergence completes over the next 24 h, and by 72 hpa, fully functional tentacles and hypostomes are formed. Basal disk upon amputation regenerates within 30–36 hpa. The morphological characteristics of a head regenerating polyp are depicted in Fig 1A (Fig 1A). Upon a screen using specific pharmacological inhibitors (Figs 1B and S1), we identified a significant role for SETD8 in head and foot regeneration. When KMT5A (SETD8) was attenuated, a significant reduction in the head regenerative ability in *Hydra* polyps was observed at all the target time points starting from 33 hpa. The emergence of tentacles is delayed by 12 h, and the polyps fail to successfully regenerate all their tentacles in the same duration as control polyps (Figs 1C and D, S2, and S3). Because SETD8 inhibition significantly impacts head regeneration, we performed a foot regeneration assay to determine the role of SETD8 in this process. After amputation of the foot, a peroxidase staining assay was employed to understand the dynamics of the regenerative process. As seen in Fig 1E, the control polyps gradually form a differentiated foot after amputation, starting from 26 hpa and completed by 36 hpa. Contrastingly, the inhibitor-treated polyps cannot regenerate the foot after amputation (Fig 1E).

## SETD8 regulates head organizer gene expression during regeneration

The wound-healing process is the first step of regeneration, and its dynamics were studied after the inhibition of SETD8. We performed staining of the F-actin filaments in the regenerating tip using phalloidin and observed the cytoskeletal structures at early time points post-decapitation. There was no significant difference in the extent of wound closure at comparable time points between the control and inhibitor-treated polyps (Fig 2A). After successful wound healing, the head organizer is formed in *Hydra*, which is needed for the differentiation of the head structures like mouth and tentacles. In *Hydra*, the Wnt signalling pathway and activity of β-catenin are required in the early stages of regeneration of both the head and the foot (Gufler et al, 2018). To understand further molecular dynamics, we performed 3′mRNA sequencing on the regenerating tips of polyps treated with UNC0379 after different regeneration times. We identified various signalling pathway members, among which, the Wnt signalling pathway components were enriched (Fig 2B). Among the various genes that are part of the Wnt signalling pathway and are direct targets of the TCF7L2 transcription factor, *Brachyury*, a bonafide head specification marker and head organizer gene, is involved in head morphogenesis and is critical for regeneration (Technau & Bode, 1999). Therefore, we used

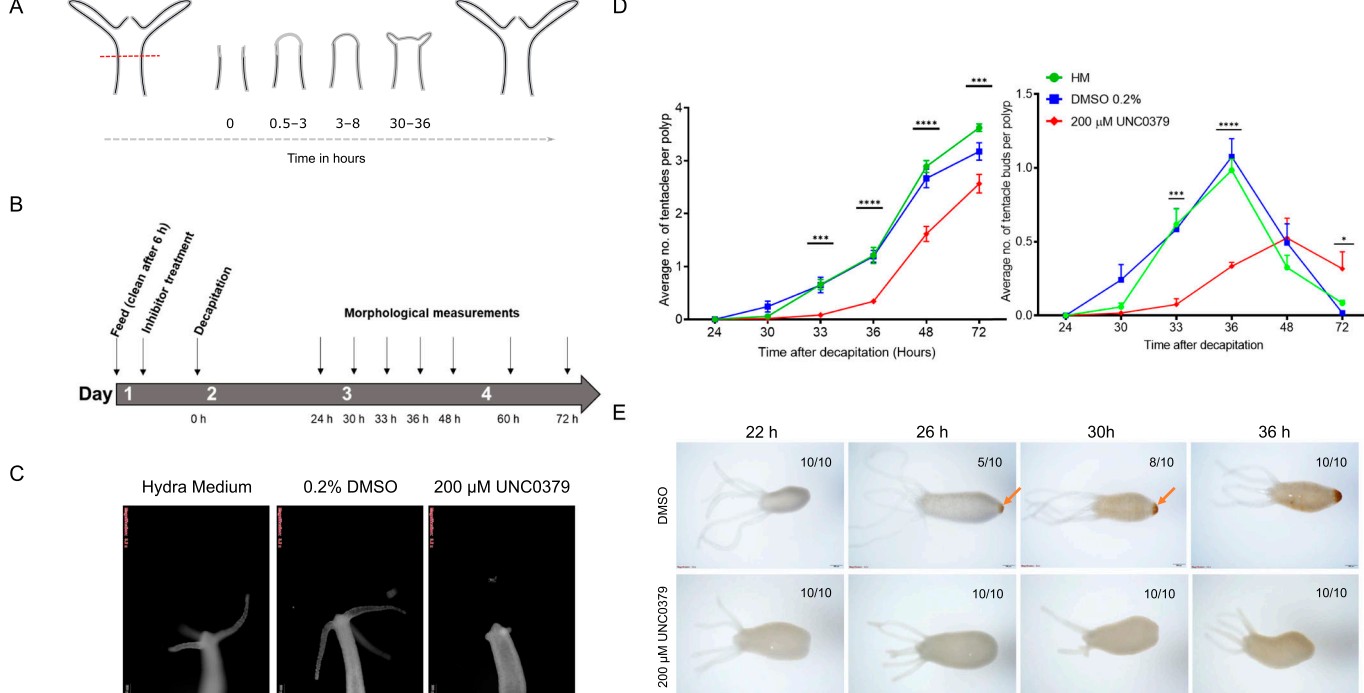

**Figure 1. SETD8 activity is necessary for regeneration in *Hydra*.**
**(A)** Kinetics of gross morphological changes during *Hydra* head regeneration. **(B)** The regime for the chemical inhibitor treatment and head regeneration assay. **(C)** Polyps decapitated and fixed at 72 hours postamputation (hpa) with and without treatment with the SETD8 inhibitor, UNC0379. **(D)** The graph depicts the average number of tentacles (left) and the number of tentacle buds per polyp (right) at each time point post-decapitation during regeneration. (N = 5, n = 25, *$P < 0.5$, ***$P < 0.001$, ****$P < 0.0001$). **(E)** A regeneration time course for observing the process of foot regeneration after amputation. Polyps fixed at 22, 26, 30 h, and 36 hpa were subjected to peroxidase-staining assay using DAB, resulting in a brown precipitate at the regenerated foot. Upon treatment with the inhibitor UNC0379, foot regeneration is severely impaired. The scale bar measures 200 $\mu$m.

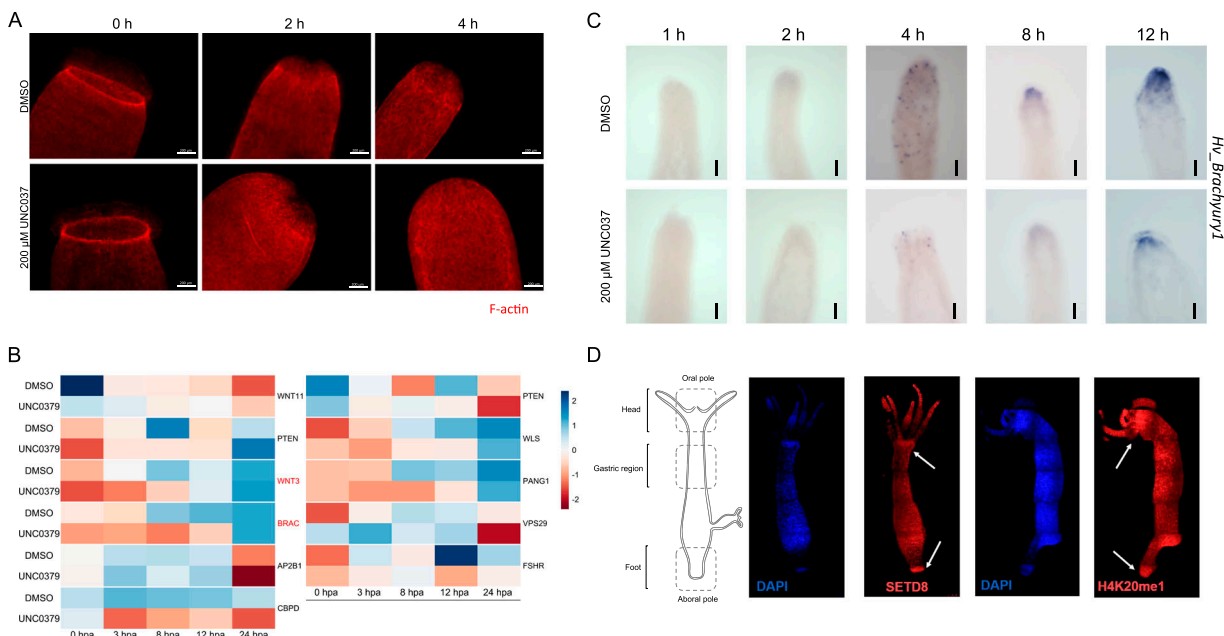

**Figure 2. SETD8 regulates head organizer gene expression during regeneration.**
**(A)** Wound healing during regeneration. The regenerating polyps at early time points are stained with phalloidin to observe the filamentous actin structures. The top row depicts control polyps, and the bottom row depicts the polyps treated with the SETD8 inhibitor. The scale bar measures 200 $\mu$m. **(B)** Differential expression of Wnt signaling-associated genes during head regeneration upon inhibition of SETD8. The heatmap depicts the expression in control and treated regenerating tips at each time point. Genes with a log$_2$ fold change cut-off of ±0.58 and a $P$-value of 0.01 have been depicted. **(C)** Whole-mount RNA in situ hybridization against the *Brachyury1* gene in regenerating polyps with and without inhibitor treatment. The blue stain depicts the expression pattern of the *Brachyury1* gene and shows that the inhibitor treatment impairs the expression of *Brachyury1*. The scale bar measures 100 $\mu$m. **(D)** The schematic of the *Hydra* polyp on the left depicts the three different regions of the polyp. The right panels depict polyps stained with $\alpha$-SETD8 and $\alpha$-H4K20me1 antibodies in red and green, respectively. The nuclei have been stained with DAPI as a counterstain.

this gene as a marker for the early morphogenetic events post-decapitation during regeneration. We performed a whole-mount RNA in situ hybridization to study the localization of the *Hv_Brachyury1* gene in the regenerating tips. In normal polyps, the expression of *Hv_Brachyury1* starts at 2 hpa at the regenerating tip. At 4 hpa, a scattered group of cells in the top 1/3$^{rd}$ of the regenerating polyp start expressing this gene which later gets clustered to the tip of the animal by 8 hpa. This focused expression pattern increases in intensity by 12 hpa and is maintained throughout the subsequent successful regeneration, as is seen in uncut normal polyps. When the polyps are treated with the SETD8 inhibitor before decapitation, the expression of *Hv_Brachyury1* is severely affected, with both the expression domains and intensity reduced at all time points of regeneration (Figs 2C and S4). To understand the localisation of SETD8 and its target histone modification in *Hydra* polyps, we performed an immunofluorescence assay using specific antibodies against the methyltransferase and the histone PTM. The occurrence of both SETD8 and its target histone modification, H4K20me1, is relatively higher in the head and the foot of the polyps (Figs 2D and S5).

### SETD8 interacts with the effector transcriptional activator of the Wnt signalling pathway in *Hydra*

*Brachyury1* is a target of the canonical Wnt/$\beta$-catenin signalling pathway in *Hydra* (Bielen et al, 2007). To understand the role SETD8 has in the patterning of the oro–aboral axis of *Hydra*, we activated the Wnt signalling pathway using Alsterpaullone (ALP). After the ALP

treatment regime shown in Fig 3A, which leads to the whole polyp turning into a head (Fig 3B), we checked the levels of SETD8 at both the mRNA and protein levels. We observed that the expression of SETD8 is up-regulated after the activation of the Wnt signalling pathway (Fig 3C). The predicted size of *Hydra* SETD8 protein is 32.2 kD, and as seen in Fig 3D, the Western blot performed with an $\alpha$-mouse-SETD8 antibody detects the target protein in the *Hydra* lysate. To further characterise the role of SETD8 in the Wnt signalling pathway, we performed a co-immunoprecipitation using an $\alpha$-$\beta$-catenin antibody. Upon probing with an $\alpha$-SETD8 antibody, we observed a clear pulldown of the SETD8 protein, indicating that SETD8 interacts with $\beta$-catenin in *Hydra* (Fig 3E). Because SETD8 interacts with and is regulated by components of the Wnt/$\beta$-catenin regulatory network, we were interested in identifying the regulation of the target histone modification, H4K20me1. In an earlier study, we identified a direct target of $\beta$-catenin named *Margin* (Reddy et al, 2019). *Margin* is a homeodomain-containing transcription factor that is up-regulated upon activation of the Wnt signalling pathway and down-regulated upon knockdown of $\beta$-catenin. Also, the promoter of *Margin* harbors binding motifs for TCF/LEF, and $\beta$-catenin binds there upon its nuclear translocation when the signalling pathway is activated (Reddy et al, 2019). Inhibition of SETD8 clearly rescues the ectopic tentacle phenotype indicating its role in regulating the Wnt signaling pathway (Fig S6). We used this experimental paradigm of ectopic activation of the Wnt signalling pathway and checked the occupancy of H4K20me1 on the promoter regions of *Margin* and *setd8*. We observed an

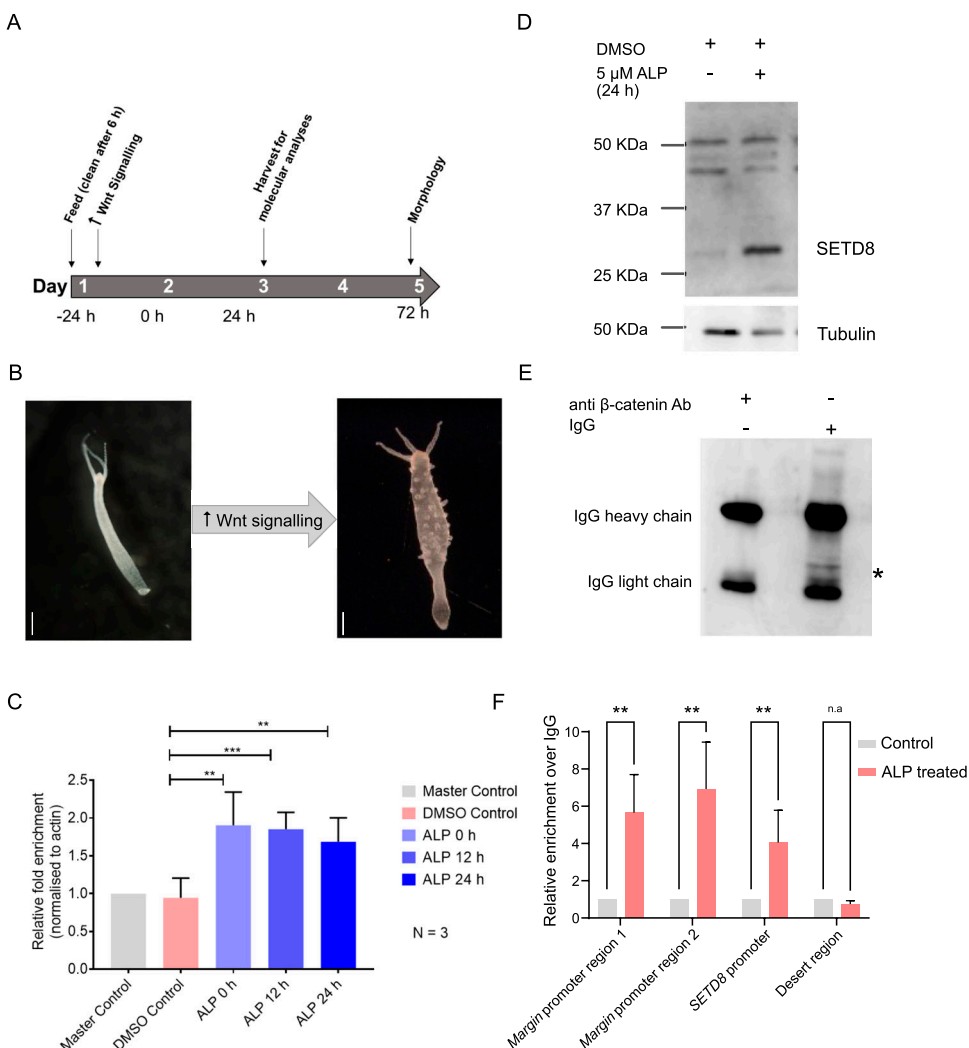

**Figure 3. SETD8 is regulated by the Wnt/β-catenin signalling pathway and is part of the transcriptional machinery with β-catenin.**
**(A)** The treatment regime used to activate the Wnt signalling pathway using ALP and study the level of SETD8. **(B)** After the activation of Wnt signalling in *Hydra*, the entire polyp turns into a head, and ectopic tentacles are seen all over the body column. **(C)** The levels of SETD8 mRNA were monitored using qRT–PCR and show that the expression of SETD8 is up-regulated after ALP treatment. **(D)** The level of SETD8 protein was monitored using Western blot, and this also shows the increased level upon activation of the Wnt signalling pathway. The size of *Hydra* SETD8 is 32.2 kD, and the molecular weight ladder is indicated on the left of the gel. **(E)** Western blot showing the co-immunoprecipitation of SETD8 upon pulldown using an α-active βcatenin antibody. The red box depicts the band corresponding to *Hydra* SETD8 protein which is 32.2 kD in size, and the asterisks mark the SETD8 band in the lysates after immunoprecipitation. **(F)** The occupancy of H4K20me1 was assayed using ChIP followed by q-PCR and plotted. The occupancy in the ALP-treated samples has been normalized to that in the DMSO control samples. (**$P = 0.0073$, Two-way ANOVA with Šídák's multiple comparisons test).

enhanced occupancy of the modification at the promoter regions of both genes (Fig 3F).

## H4K20me1 occupancy is context-dependent and is linked to specific signalling pathways

To elucidate the role of this unique modification in the regeneration of *Hydra*, we performed ChIP-sequencing for H4K20me1 in Wnt-activated polyps (Fig S7) and regenerating tips at five time points (Fig S8) and identified the differentially occupied peaks relative to DMSO-treated controls and 0 hpa regenerating tips, respectively (Figs 4 and 5). The occupancy of H4K20me1 was plotted around the peak centres at the differentially occupied genomic regions (Figs 4A, 5A–C, and 6A–D). K-means clustering revealed four clusters of gene bodies in the *Hydra* genome based on the occupancy of H4K20me1 across the compared regions under both physiological conditions. The clusters depict how the occupancy of H4K20me1 changes. The genes associated with the differentially occupied genomic regions were used to perform GO enrichment and interaction analyses using STRING. Upon ALP treatment and

investigation of differentially H4K20-methylated regions, we identified a limited number (159) of genomic regions that form four clusters based on the occupancy of H4K20me1. Whereas clusters 1 and 3 display an increase in H4K20me1 occupancy upon activation of Wnt signalling, clusters 2 and 4 display a decrease (Fig 4A). Under this physiological condition, we observe a positive correlation between H4K20me1 occupancy and transcription. The genes in cluster 1 code for housekeeping proteins such as RNA polymerases and proteins, including cell ubiquitin ligases involved in catabolic processes (Fig 4C). Their transcription is modestly increased upon ALP treatment, and the occupancy of H4K20me1 is also higher than in control polyps. Cluster 3 is of great interest because it shows a highly interconnected network of Wnt signalling-related proteins acting both in the canonical and non-canonical arms of the signalling pathway. Furthermore, proteins involved in extensive morphogenesis by altering cellular shape are also found in this cluster (Fig 4E). Their transcription is highly up-regulated upon activation of Wnt signalling, with a higher occurrence of H4K20me1 correlated to it. Cluster 2 harbors genes that exhibit decreased expression upon ALP treatment. Genes involved in the proliferation

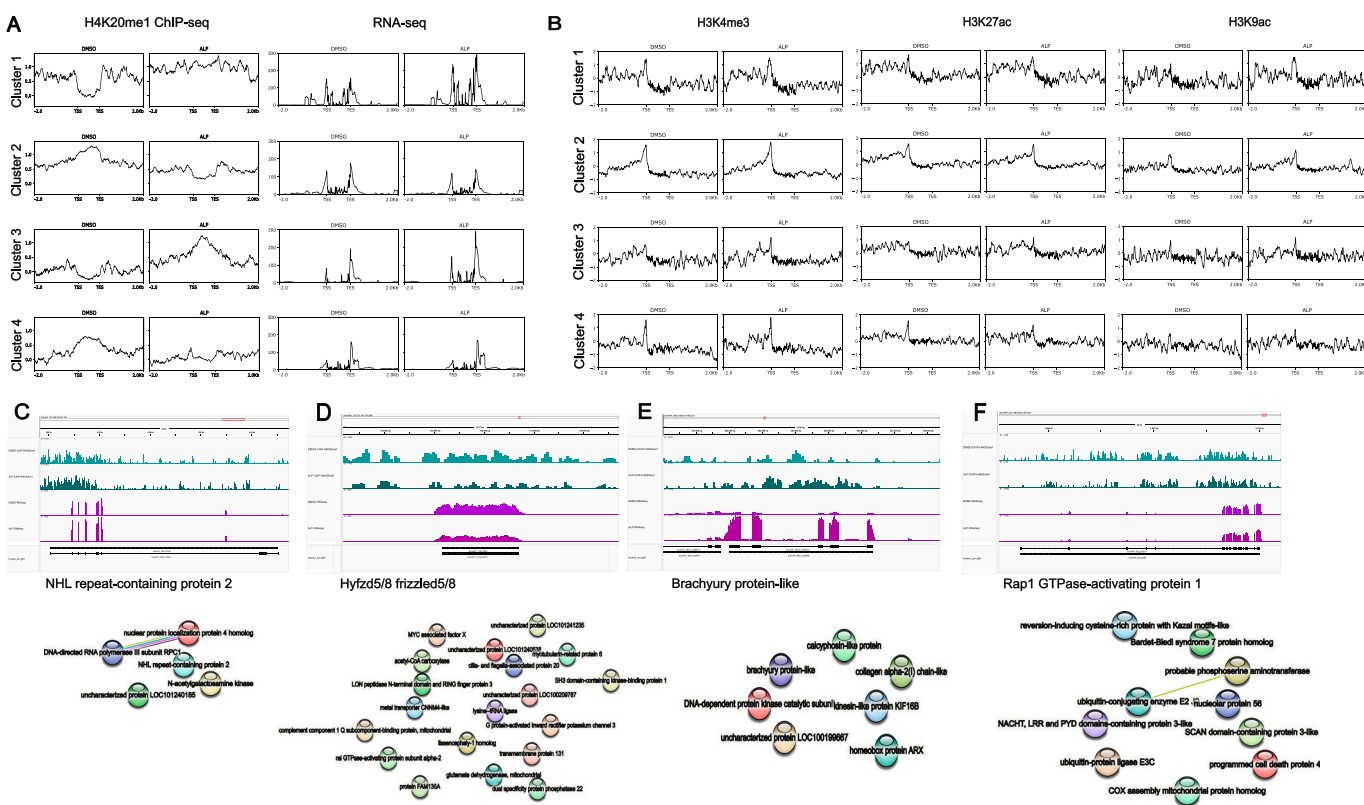

**Figure 4. Differential occupancy of H4K20me1 upon ectopic activation of the Wnt signalling pathway is associated with limited gene sets.**
**(A)** The differential occupancy of H4K20me1 across four clusters upon ALP treatment in *Hydra vulgaris* Ind-Pune. The four clusters with differential peaks of H4K20me1 are correlated to the RNA-seq data across the nearby transcripts. Profiles suggest how H4K20me1 has a dual role in transcription activation and repression. **(B)** The four clusters obtained based on the occupancy of H4K20me1 on gene bodies were used to investigate the occupancy of the three activation-associated histone marks: H3K4me2, H3K4me3, and H3K27ac in ALP-treated and control polyps. **(C, D, E, F)** The top panels show Integrated Genome Viewer screenshots of the occupancy and transcript abundance of representative loci from clusters 1–4, respectively. The names of the respective loci are indicated below the Integrated Genome Viewer screenshots. The bottom panels depict the STRING interaction analysis of genes associated with differentially methylated genomic regions of clusters 1–4, respectively, performed using protein names from *Hydra vulgaris* genome version 3 annotation.

of cells, including Myc and the Wnt receptor, Frizzled, are found in this cluster (Fig 4D). The expression of these genes decreases upon ectopic Wnt activation with a corresponding decrease in H4K20me1. Although cluster 4 shows a decrease in H4K20me1 occupancy upon ALP treatment, there is no alteration of gene expression, and a few genes involved in ubiquitination and apoptosis are part of this cluster (Fig 4F). However, at the 159 differentially occupied regions for H4K20me1, although there is a change in gene expression, the corresponding activation-associated histone marks do not display any significant alterations (Fig 4B). To understand the transcriptional regulation of the genes associated with the four genomic region clusters, we plotted the logCPM values (Figs S9 and S10). The gene expression corroborated the average profiles of the RNA-seq read distribution. Transcription factors of the Wnt signalling pathway, found in the third cluster, exhibit increased expression upon ALP treatment as expected (Fig S10).

Contrastingly, upon retrieval of differentially occupied regions during the regeneration time course, we found a large number of genomic regions with altered H4K20 methylation status compared with the 0 h time point (4,897, 5,116, 5,657, 5,400 at 3, 8, 12, and 24 hpa respectively). These regions again form four k-means clusters based on the trend displayed by the occupancy of H4K20me1. The

genes associated with the differentially regulated peaks were retrieved, and a STRING interaction analysis was performed to identify the interactions between the genes (Figs S12–S24). Across time points, cluster 1 denotes regions with visibly increased occupancy of H4K20me1 (Figs 5A–C and 6A–D), and cluster 2 denotes those with highly decreased H4K20me1 occupancy (Figs 5A–C and 6A–D). Cluster 3 contains regions with a slight reduction in the occupancy (Figs 5A–C and 6A–D), and cluster 4 genes exhibit no visible change, although the peaks have been identified to be differentially occupied statistically (Figs 5A–C and 6A–D). Interestingly, across all time points, we observed a decrease in the occupancy of the activation-associated histone marks relative to the 0 h time point of regeneration (Fig 5D–L). The readout of active transcription is the resulting gene expression, which is mediated by opening compacted chromatin. The available ATAC-seq (Assay for Transposase-Accessible Chromatin using sequencing) data enabled investigation into the chromatin accessibility during regeneration (Cazet et al, 2021), and the RNA sequencing data are the final readout for gene regulation in both ALP-treated conditions (in-house) and regenerating tips (Murad et al, 2021). Although we did not find a common theme or genes belonging to a few known signalling pathways enriched in specific clusters (Figs S11–S26), we noted many epigenetic modifier

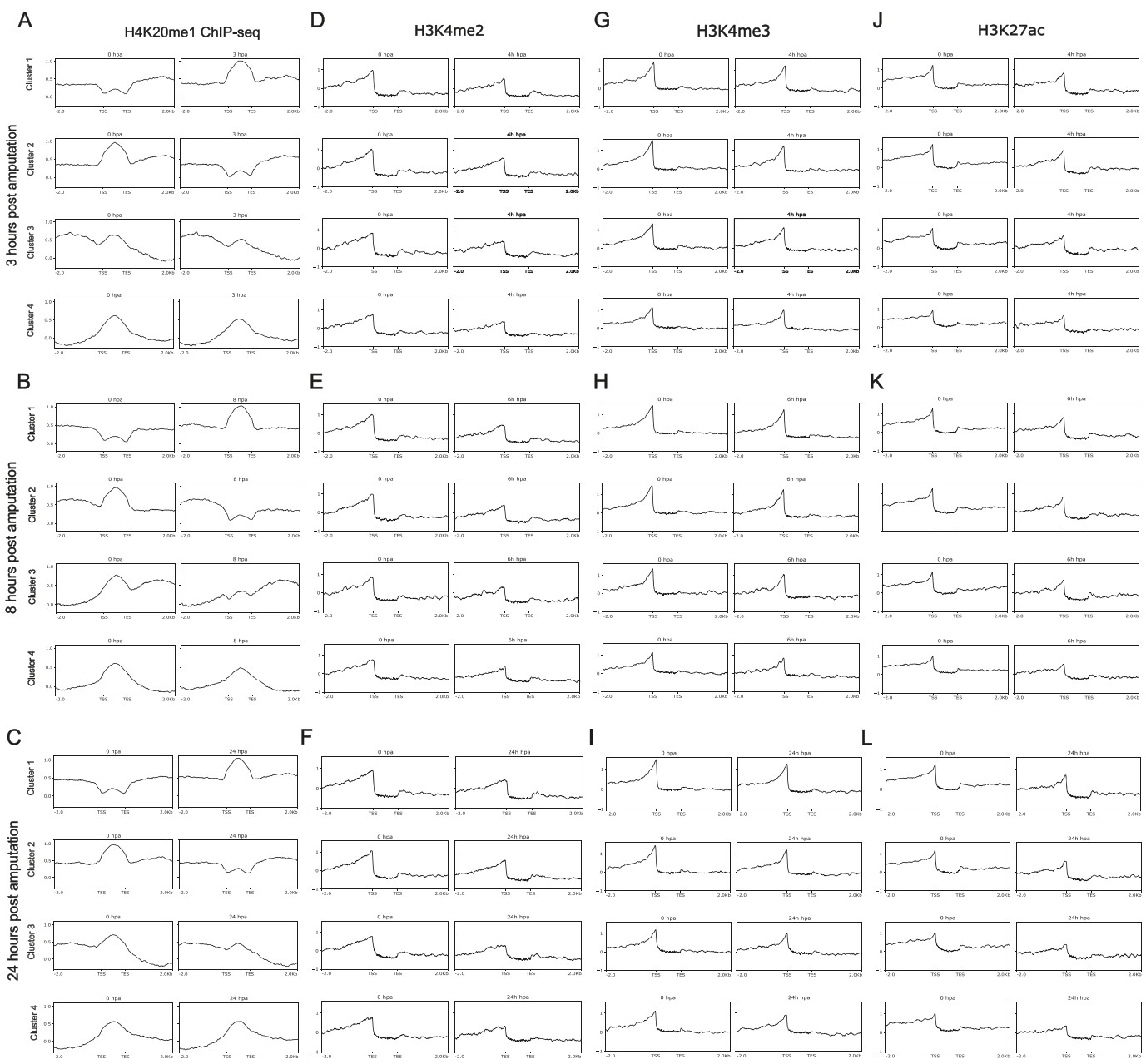

**Figure 5. Extensive differential occupancy of H4K20me1 during head regeneration indicates a putative role in the regulation of diverse processes in *Hydra*.**
**(A, B, C)** The occupancy of H4K20me1 on differentially occupied gene bodies across different regenerating time points after head amputation. **(D, E, F, G, H, I, J, K, L)** This occupancy has been plotted along with ChIP-seq data for activation-associated histone marks H3K4me2 (D, E, F), H3K4me2 (G, H, I), and H3K27ac (J, K, L). Available ChIP-seq data (Murad et al, 2021) for H3K4me3, H3K27ac, and H3K4me2 are plotted beside the H4K20me1 profiles for comparison and establishment of the role in transcriptional regulation.

genes regulated by differential occupancy of H4K20me1. How-ever, upon analysing the openness of chromatin and tran-scriptional outputs, we observed that, upon UNC treatment, transcription is increased in those genes that display higher levels of H4K20me1 alteration (Fig 6E–K). We further zoomed into a few genes to monitor the occupancy profile of this histone mark. As seen in the integrated genome viewer screenshots (Fig 6L–O), the occupancy of H4K20me1 extends beyond the pro-moter regions into the exons and introns of the gene bodies.

The differential occupancy presumably results from dynamic changes in the location of H4K20me1 across the gene body. To understand the transcriptional regulation of the genes asso-ciated with the four genomic region clusters, we plotted the logCPM values of the RNA-seq data from regenerating tips with and without inhibition of SETD8 (Figs S27–S29). Supplementing the average gene profiles, the individual gene expression profiles of the Wnt pathway-associated genes demonstrate how organizer formation is disrupted when SETD8 is inhibited. The plots depict the gene expression at each

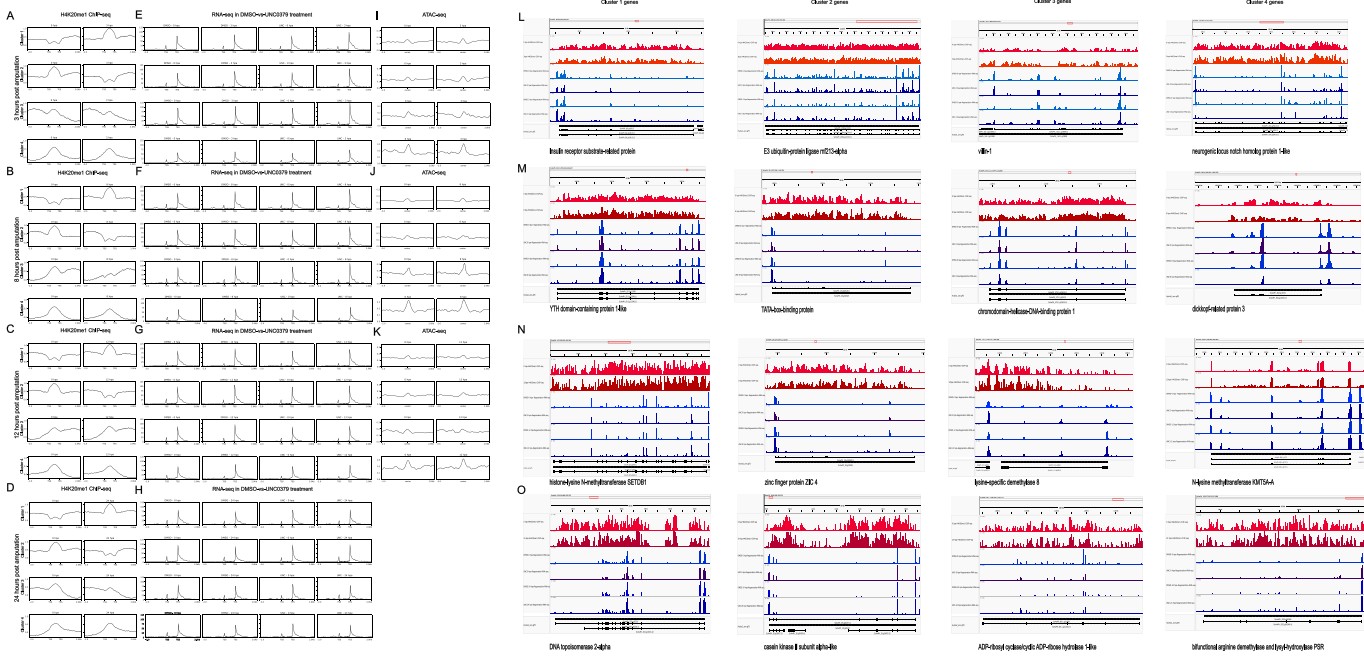

**Figure 6. Transcriptional regulation by differentially methylated H4K20me1 during head regeneration in *Hydra*.**
**(A, B, C, D)** The four clusters obtained based on the occupancy of H4K20me1 on differentially methylated gene bodies at 3, 8, 12, and 24 hpa, respectively, have been used to understand chromatin accessibility and the transcription of the respective genes. **(E, F, G, H)** The RNA-seq reads in control and UNC0379-treated regenerating tips at 3, 8, 12, and 24 hpa, respectively, have been plotted for the four clusters separately and represent the resultant transcription from those regions. **(I, J, K)** The differentially methylated peaks have been correlated to ATAC-seq data at the same time points as Cazet et al (2021). **(L)** IGV snapshots for representative loci in clusters 1–4 of the differentially methylated regions at 3 hpa. **(M)** IGV snapshots for representative loci in clusters 1–4 of the differentially methylated regions at 8 hpa. **(N)** IGV snapshots for representative loci in clusters 1–4 of the differentially methylated regions at 3 hpa. **(O)** IGV snapshots for representative loci in clusters 1–4 of the differentially methylated regions at 3 hpa.

regenerating time point in the presence and absence of SETD8 inhibition. SETD8 inhibition disrupted the expected expression patterns of genes. At different time points, the effect is different, underscoring the requirement of SETD8 in various transcriptional networks (Figs S28 and S29).

## Discussion

Regeneration recapitulates developmental patterning and is aided by extensive changes in chromatin architecture and, consequently, gene expression (Fig 1A). Various epigenetic modifiers regulate these changes; hence, we sought to identify one such class of methylome modulators in *Hydra* axis patterning and regeneration. Upon chemical inhibition, we identified a significant role for the histone methyltransferase SETD8 in both head and foot regeneration (Figs 1C–E and S1). Inhibition of SETD8 led to a transient reduction of the target H4K20me1 histone modification, which resulted in delayed head regeneration (Fig S2). The emergence of tentacles is delayed by 12 h, and the polyps cannot complete the differentiation of all the tentacles within 72 h. Foot differentiation is also severely impacted after the chemical treatment. In *Hydra*, the Wnt signalling pathway and activity of β-catenin are required in the early stages of regeneration of both the head and the foot (Gufler et al, 2018). We investigated the process of head regeneration in deeper detail to understand the mechanism of SETD8 function in

*Hydra* physiology. The earliest step of regeneration, wound healing, is morphologically not affected by the inhibition of SETD8 (Fig 2A). However, at a molecular level, various Wnt signalling components, apoptosis-related genes, Jun kinase, and many insulin signalling components are dysregulated upon inhibition of SETD8 (Fig 2B). We attempted to genetically inhibit the function using siRNA-mediated knockdowns. However, unlike transcription factor knockdowns that are very stable (Reddy et al, 2019), the knockdown of epigenetic regulators, specifically histone methyltransferases, is very transient, and the feedback upregulation does not allow a sufficient reduction in the enzyme expression to affect subsequent morphogenetic events. Regeneration assays performed after knockdown suggest the same, wherein the effect on regeneration is less pronounced than upon chemical inhibition (Fig S30). We are aware of the compounding effects of inhibiting SETD8 on its non-histone substrates, nevertheless, we still attribute this to the effects on catalytic function because of the results we obtained using genetic inhibition.

Brachyury is a bonafide head specification marker among the various genes that are part of the Wnt signalling pathway and direct targets of the TCF7L2 transcription factor. The formation of the head organizer by expression of various patterning transcription factors like *Brachyury* is dramatically altered when SETD8 is inhibited (Figs 2C and S4). Impairment of both the head and foot regeneration by inhibiting the SETD8 enzyme indicates a role in the position-independent function of the Wnt signalling pathway. SETD8 in *Hydra* is up-regulated at both mRNA and protein levels upon

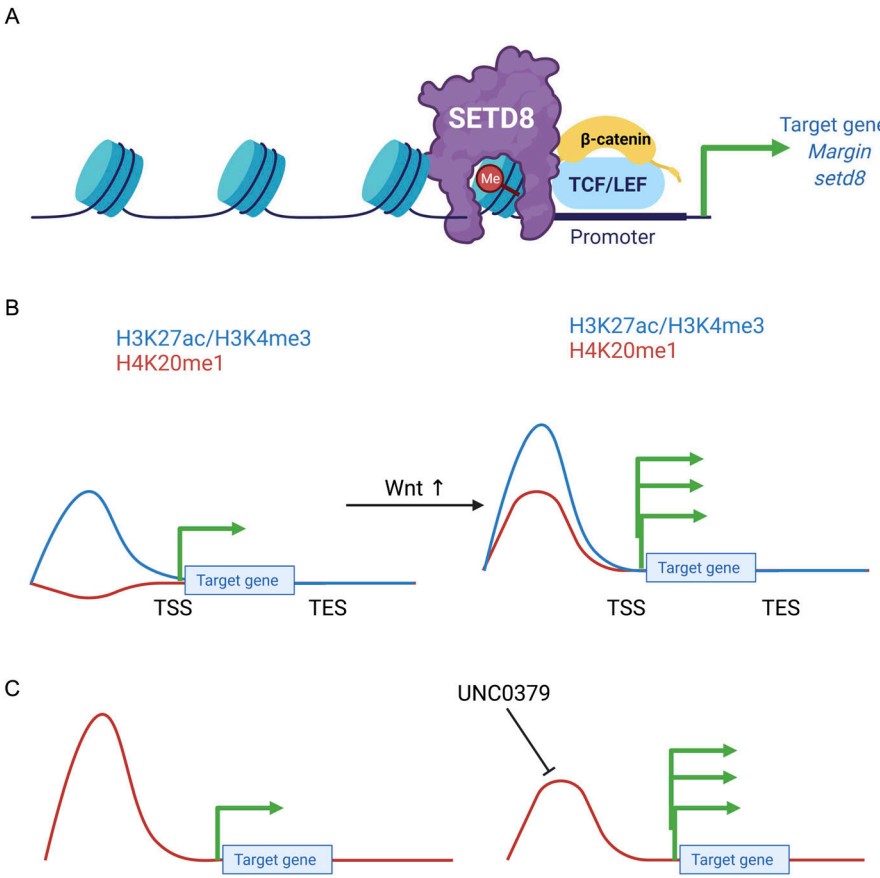

**Figure 7. Molecular mechanism of the action of SETD8 and H4K20me1 in transcriptional regulation of *Hydra* genes.**
**(A)** Upon activation of the Wnt signalling pathway, β-catenin is translocated to the nucleus, where it interacts with the TCF7L2 transcription factor to activate the transcription of target genes. The effective activation of the signaling pathway is correlated with SETD8 interacting with β-catenin at the promoters of the target genes and depositing the H4K20me1 mark on the nucleosomes. **(B)** Upon ectopic activation of Wnt signalling, based on our identification of differentially methylated regions, we observe a positive correlation between H4K20me1 and activation of transcription. **(C)** Based on our analysis of differentially methylated regions during regeneration, we contrastingly observe a predominantly negative correlation with transcription. Upon loss of H4K20me1 on various differentially methylated regions, the associated genes exhibit enhanced transcription.

activating the Wnt signalling pathway and physically interacts with β-catenin in the nucleus (Fig 3C–E). The activity of SETD8 is necessary for activating genes down-stream of the Wnt/β-catenin signalling pathway (Fig S6). This indicates a conserved mode of action for SETD8 (Li et al, 2009; Huang et al, 2021) in *Hydra*, wherein the axis patterning processes are continuously active and play a role in transcriptional regulation.

To decipher the molecular mechanisms underlying transcriptional regulation by SETD8, we characterized the target histone modification H4K20me1. Antibodies against the modification on the completely identical histone H4 were used to validate the presence of H4K20me1 in *Hydra*, allowing us to study its dynamics. H4K20me1 is present all across the body column of *Hydra* but has a slightly higher localization near the hypostome and the foot of the polyp (Figs 2D and S5). These are two regions of the polyp with greater numbers of differentiating cells and the most patterning events taking place. The higher presence of the histone mark indicates a putative role in these processes. When the body axis is perturbed by ectopic activation of the Wnt signalling pathway, the whole polyp acquires head-like characteristics and tentacles arise all over the body column. In these conditions, we identified an enhanced occupancy of H4K20me1 at the promoter regions of a homeodomain transcription factor *Margin*, an established direct target of β-catenin (Reddy et al, 2019). In addition, the regulation of *setd8* expression is also associated with the presence of H4K20me1 at its

promoter, indicating a self-regulatory role for SETD8 downstream of the Wnt signalling pathway (Fig 3F). We have identified the molecular mechanism of action of SETD8 wherein, upon switching on the Wnt signalling pathway, β-catenin translocates into the nucleus. This leads to the interaction of SETD8 and β-catenin at the TCF-binding regions of the genome and culminates in transcriptional activation, as seen for both the validated targets *Margin* and *setd8* and morphogenesis (Fig 7A).

Investigation of the steady-state global occupancy of H4K20me1 upon ectopic activation of the Wnt signalling pathway revealed that H4K20me1 excludes sites occupied by the transcription activation-associated histone marks described in a previous study (Reddy et al, 2020). The occupancy of H4K20me1 is very low globally because of it being a primary histone mark and the consecutive di and trimethyl marks being part of constitutive heterochromatin. When the axis of the organism is perturbed by the activation of a single signalling pathway, the number of differentially occupied genomic regions is limited (Fig 4). Genes involved in cell proliferation, differentiation, patterning, apoptosis, and Wnt signalling are among the differentially methylated regions in the genome (Fig 4C–F). When a physiological process such as regeneration is activated, it results in more large-scale changes in gene expression of different signalling pathways, further reflecting at the epigenetic level. We also observe the same with the occupancy of H4K20me1, which is altered along a large portion of the *Hydra* genome (Fig 5). H4K20me1

is a broad occupancy histone mark with extensive dynamics. The occupancy of H4K20me1 appears to regulate transcription based on its location on the genome. Loss of SETD8 enzyme activity increases transcription at genes that are most susceptible to changes in H4K20me1 occupancy (clusters 1, 2 in Fig 6E–H), indicating a predominantly negative correlation of H4K20me1 with transcription. The occurrence of H4K20me1 extends downstream of the promoters and into gene bodies (Fig 6L–O), indicating an association with transcriptional elongation. H4K20me1 occupancy is known to be associated with the elongation of rapidly transcribing genes in mammalian cells (Vakoc et al, 2006; Veloso et al, 2014). Contrastingly, in senescent cells and neurons, H4K20me1 occupancy seems to prevent elongation (Wang et al, 2015; Tanaka et al, 2017). Thus, the effect of H4K20me1 occupancy on transcription appears to be highly context-specific and regulated in a cell type-specific and location-specific manner.

The role of SETD8 and H4K20me1 in transcriptional regulation is not well established. At promoters of genes involved in erythropoiesis, loss of SETD8 causes reduced chromatin accessibility and impaired differentiation (Myers et al, 2020). In zebrafish and *Drosophila*, H4K20me1 is positively correlated with transcription in a few developmental contexts involving the Wnt signalling pathway (Li et al, 2011; Huang et al, 2021). This has also been observed in global high-resolution profiling done in human T cells (Barski et al, 2007). We observe this positive correlation with transcriptional regulation in the Wnt/$\beta$-catenin signalling network in *Hydra*, and the delay in regeneration can be explained by a reduction specifically in the Wnt-mediated differentiation potential of the *Hydra* cells to regenerate lost head and foot structures. A contrasting role for this modification has also been identified using in vitro studies involving the L3MBT1 protein, which is critical for transcriptional repression (Kalakonda et al, 2008). During the progression of cell cycle stages, the SETD8 protein is necessary for maintaining chromatin condensation and acts via the PCNA axis (Abbas et al, 2010). This is important to maintain the repressed state of transcription to allow mitosis to move forward and is additionally facilitated by the reduction in the levels of the demethylase PHF8 in the prophase of cell division (Karachentsev et al, 2005; Liu et al, 2010). However, in the context of a complex process such as *Hydra* head regeneration, where there is a necessity for various signalling pathways to interact, the trend indicates that H4K20me1 is negatively correlated with the transcription of genes. The regeneration process involves multiple cellular processes, one of which is the reduced cellular proliferation and increased transdifferentiation in the morphallactic regeneration of *Hydra*. This is also evident from the transcriptional regulation observed upon activation of only a single pathway (Figs S9 and S10) in contrast to an entire physiological process being perturbed (Figs S28 and S29). There is also a clear distinction in the genomic regions that show differential occupancy in the ALP-treated and the regenerating polyps indicating a diverse role for the histone mark in transcriptional regulation (Fig S31). Therefore, the H4K20me1 mark at a global scale may show an association with reduced transcription, similar to its role in cell cycle regulation. Whereas SETD8 is conserved in flies and humans, the enzyme and the target H4K20me1 have distinct roles in the eye development of *Drosophila* (Crain et al, 2022). The transient decrease in the target mark in *Hydra* and the marked effect on

regeneration observed also point towards such a role in the Cnidarian.

Therefore, this mark's dual mode of action is more specific than we previously could decipher. The specificity could be at TF-binding sites, as seen with ectopic activation of the Wnt signalling pathway (Fig 4), and a deeper investigation of the TCF-binding sites displayed distinct regulation of H4K20me1 at these motifs in response to Wnt signalling activation (Fig 3F). We have, therefore, identified a putative repressive mark in *Hydra* downstream of the H4K20me1 mark. The typically pericentric H4K20me2 and me3 marks may have a transcriptional regulatory role in this early eumetazoan which evolved to be restricted only to pericentromeric heterochromatinization in higher animals. The unique physiology of *Hydra* enables studying and visualising the different modes of action for the histone PTM, H4K20me1, and understanding its dichotomous nature in transcriptional regulation (Fig 7B and C).

# Materials and Methods

## *Hydra* culture

A clonal culture of *Hydra vulgaris* Ind-Pune was maintained at 18°C in Hydra medium (HM) (100 $\mu$M KCl, 100 $\mu$M MgSO$_4$.7H$_2$O, 1 mM CaCl$_2$.2H$_2$O, 1 mM Tris (pH-8.0), 1 mM NaCl) using standard protocols described previously (Horibata et al, 2004). *Hydra* polyps were fed with freshly hatched *Artemia nauplii* larvae daily and cleaned 6–8 h post-feeding.

## Inhibitor treatments

To perform the inhibitor treatments, the following inhibitors were used: Pinometostat (EPZ5676) for the KMT4 enzyme (Cat # S7062; Selleckchem), MM-102 for the KMT2A enzyme (Cat # S7265; Selleckchem), UNC0379 for the KMT5A enzyme (Cat # S7570; Selleckchem; Cat # 16400; Cayman Chemicals Co.), PFI-2 HCl for the KMT7 enzyme (Cat # S7294; Selleckchem), GSK-J4 for the enzyme KDM6A/6B (Cat # SML0701-5MG; Sigma-Aldrich), GSK-LSD1 for the enzyme KDM1A (Cat # SML1072-5MG; Sigma-Aldrich), and GSK343 for the enzyme KMT6A (Cat # SML0766; Sigma-Aldrich). To identify the working concentration of each inhibitor, 50 *Hydra* polyps were treated for 8 h (overnight), and the concentration at which 50% of the polyps were disintegrated was identified (LD$_{50}$). A concentration below the LD$_{50}$ was used to perform the regeneration assays.

## Head regeneration assay

*Hydra* polyps were treated with target inhibitors for 8 h and decapitated. After decapitation, the inhibitor treatment was continued till 72 h with fresh addition of inhibitor-containing Hydra medium every 24 h. The number of tentacle buds and tentacles was manually counted and recorded at 24, 30, 33, 36, 48, and 72 h. The statistical analysis was performed using GraphPad Prism version 7.0c for MacOS (GraphPad Software, www.graphpad.com).

## Foot staining assay

*Hydra* polyps were treated with target inhibitors for 8 h, and the foot was amputated. After amputation, the inhibitor treatment was continued till 36 h with fresh addition of inhibitor-containing Hydra medium every 24 h. At the target time points of 22, 26, 30, and 36 hpa, the polyps were relaxed with 2% urethane/HM for 2 min and fixed using 4% PFA/PBS for 1 h at room temperature. After fixation, specimens were given three 10-min washes in 1X PBS + 0.25% Triton X-100 (PBST). The animals were then incubated for 15 min at room temperature in 1X PBST containing 0.02% diaminobenzidine (DAB) and 0.003% hydrogen peroxide ($H_2O_2$). After incubation, they were rinsed for 30 min in PBST, mounted on slides in phosphate-buffered glycerol, and examined with a stereo zoom microscope (Zeiss).

## Whole-mount in situ hybridization on regenerating tips

*Hydra* polyps were treated with target inhibitors for 8 h and decapitated. After decapitation, the inhibitor treatment continued until the target time points of 1, 2, 4, and 8 hpa. The polyps were then relaxed by treatment with 2% urethane/HM for 2 min and fixed using 4% PFA/HM at 4°C overnight. Digoxigenin-labelled RNA probes for *Hv_Bra1* were prepared by in vitro transcriptions from templates amplified from a recombinant pCR Blunt II TOPO (Cat # 450031; Invitrogen) plasmid containing the *Hv_Bra1* gene using PCR. (DIG Labelling Mix, Cat # 1277073910; Sigma-Aldrich; SP6 RNA Polymerase, Cat # 10810274001; Sigma-Aldrich; T7 RNA Polymerase, Cat # 10881767001; Sigma-Aldrich). Whole-mount in situ hybridization was performed on the polyps as described previously (Martinez et al, 1997) with the following changes. Treatment with proteinase-K was performed for 5 min, and heat inactivation of the endogenous alkaline phosphatases was done at 70°C for 15 min in 1X SSC. Digoxigenin-labelled RNA probe at a concentration of 150 ng/ml was used for hybridization at 59°C. The post-hybridization washes were performed using 1X SSC-HS gradients. After staining with 50% NTMT/50% BM-purple AP substrate for 1 h at room temperature, the animals were mounted in 80% glycerol for imaging.

## Actin filament staining

*Hydra* polyps were treated with target inhibitors for 8 h and decapitated. After decapitation, the inhibitor treatment was continued till the target time points of 0, 2, and 4 hpa. At the target time points, the polyps were relaxed with 2% urethane/HM for 2 min and fixed using 4% PFA/PBS for 1 h at room temperature. After three 10-min washes in 1X PBS and permeabilization in 1X PBS + 0.1% Triton X-100 (1X PBST), staining was done with Alexa Fluor 568 phalloidin (Cat # A12380; Invitrogen) diluted 1:200 in 1X PBST at room temperature for 1 h. Finally, the polyps were washed in 1X PBST (3 × 10 min), mounted on slides using VECTASHIELD Antifade Mounting Medium (Cat # H-1000-NB; Novus Biologicals) and imaged using the ApoTome microscope (Carl Zeiss).

## Immunofluorescence assay

*Hydra* polyps were treated with target inhibitors for 8 h and decapitated. After decapitation, the inhibitor treatment was continued till the target time points of 0, 2, and 4 hpa. At the target time points, the polyps were relaxed with 2% urethane/HM for 2 min and fixed using 4% PFA/PBS for 1 h at room temperature. After fixation, the polyps were permeabilized using 1X PBS + 0.1% Triton X-100 (1X PBST) for 30 min with changes every 10 min. Blocking was done with 1X PBST + 20% FBS for 1 h at 4°C. The solution was replaced with a fresh blocking solution containing *α*-H4K20me1 (rabbit polyclonal IgG, Cat # CS200569; EMD Millipore) primary antibody at a dilution of 1:100 and incubated overnight at 4°C. After washing with 1X PBST thrice for 10 min each, the polyps were incubated in a solution containing a secondary antibody at a dilution of 1:100 (anti-rabbit IgG-Alexa568, Cat # A11011; Invitrogen). After washing, the nuclei were stained with 0.5 *µ*g/ml DAPI for 10 min at room temperature. Finally, the polyps were washed in 1X PBS thrice for 10 min each, mounted on glass slides using Vectashield mounting medium, and imaged using the ApoTome microscope (Carl Zeiss).

## Co-immunoprecipitation

To perform co-immunoprecipitation, 100 polyps were collected and washed thrice with Hydra medium. The medium was removed entirely, and the polyps were resuspended in RIPA buffer (50 mM Tris–HCl pH-7.4, 150 mM NaCl, 2 mM EDTA, 0.1% vol/vol SDS, 1% vol/vol NP-40, 0.5% wt/vol sodium deoxycholate, 30 mM sodium fluoride, 0.2 mM sodium orthovanadate) and incubated on ice for 30 min. The sample was centrifuged at 4°C and 23,000*g* for 20 min twice, and the supernatant was collected. The amount of protein was quantitated using the Pierce BCA Protein Assay Kit. Five micrograms of *α*-active-*β*-catenin antibody (Cat # 05-665; Sigma-Aldrich) was added to 700 *µ*g of *Hydra* lysate and incubated at 4°C overnight. An equal amount of the appropriate IgG control was also used (Normal mouse IgG, Cat # 12-371; Sigma-Aldrich). 5% or 35 *µ*g of protein was used as the input control. After pull down, the samples were incubated with 20 *µ*l of Dynabeads M-280 sheep anti-mouse IgG (Cat # 11202D; Invitrogen) for 3 h at 4°C with gentle inverting. The supernatant was discarded, and the beads were washed with RIPA buffer thrice for 10 min each. The beads were then heated in 1X Laemmli buffer at 95°C for 15 min to denature and retrieve all the bound proteins. This was separated on an SDS–PAGE gel and subjected to Western blot using the *α*-SETD8 antibody (Anti-SETD8 [hPR-SET7] Antibody, Cat # 06-134; Sigma-Aldrich).

## Acid extraction of histones

Two hundred *Hydra* polyps were homogenized in lysis buffer or wash buffer (250 mM sucrose, 50 mM Tris-Cl, 25 mM KCl, 5 mM MgCl$_2$, 0.2 mM PMSF, 50 mM NaHCO$_3$, 0.2% Triton X-100, 45 mM Na-Butyrate, 10 mM *β*-mercaptoethanol, 1X PIC) and nuclei isolated by centrifugation at 800*g* for 15 min at 4°C. Histone proteins were isolated using the acid extraction method and resolved by 18% SDS–PAGE as previously described (Jayani et al, 2010). Briefly, the nuclear pellet was resuspended entirely in 600 *µ*l of 1 $NH_2SO_4$ and incubated for 2 h at 4°C. Histones were precipitated overnight at –20°C from the supernatant by adding trichloroacetic acid to a final concentration of 33%. Chilled acetone washes were carried out to avoid carryover of trichloroacetic acid. Histone pellets were air-dried, resuspended in 1x PBS, and resolved by SDS–PAGE.

## RNA sequencing

The polyps were decapitated, and the regenerating tips were collected at 0, 3, 8, 12, and 24 hpa. RNA was isolated using Trizol and used to perform 3' mRNA sequencing using the QuantSeq 3' mRNA-Seq Library Prep Kit FWD for Illumina (Lexogen) according to the manufacturer's instructions. The libraries were loaded onto a NextSeq 500/550 High-Output v2.5 Kit (75 cycles) (Cat # 20024906; Illumina) and sequenced using 35 cycles for read1, 8 cycles for indexes, and 35 cycles for read2 on Nextseq 550 (Illumina) at the sequencing facility of IISER Pune. The PCA and volcano plots for the 3' RNA-seq data are shown in Fig S25.

## Chromatin immunoprecipitation (ChIP)

Two thousand *Hydra* polyps per time point were decapitated, and the regenerating tips were collected at 0, 3, 8, 12, and 24 hpa to perform ChIP. 2,000 *Hydra* polyps per treatment condition were used for ALP treatment and fixed for ChIP. The *Hydra* polyps and the regenerating tips were collected and cross-linked with 1% methanol-free formaldehyde. The fixation reaction was quenched with 150 mM glycine. The fixed regenerating tips were resuspended in a Swelling Buffer (25 mM Tris–HCl pH 7.9, 1.5 mM MgCl$_2$, 10 mM KCl, 0.1% NP-40, 1 mM DTT, 0.5 mM PMSF, 1X Protease inhibitor cocktail/PIC) and homogenized using a Dounce homogenizer (35 strokes) to lyse the cells and release the nuclei. The nuclei were pelleted, resuspended in sonication buffer (50 mM Tris–HCl pH 7.9, 140 mM NaCl, 1 mM EDTA, 1% Triton X-100, 1% SDS, 0.1% Sodium deoxycholate, 0.5 mM PMSF, 1X PIC), and incubated for 30 min on ice. They were then sonicated to obtain an average chromatin size of 300 bp. Chromatin was pre-cleared using 50 $\mu$l of a 50% protein A sepharose (GE Healthcare) slurry for 1 h at 4°C with gentle inverting. Immunoprecipitations were carried out in the ChIP buffer (16.7 mM Tris–HCl pH 8.0, 167 mM NaCl, 1.2 mM EDTA, 1.1% Triton-X 100, 0.01% SDS, 1X PIC) with Anti-H4K20me1. An appropriate IgG control was also used with inverting at 4°C for 14–16 h. The samples were then incubated with 50 $\mu$l of a 50% Protein A sepharose slurry (saturated with 0.5% BSA and 10 mg/ml yeast tRNA) for 3 h at 4°C with gentle inverting. ChIP samples were reverse-cross-linked, and the DNA was purified using the phenol:chloroform:isoamyl alcohol-based precipitation method. Input chromatin was obtained after pre-clearing by de-cross-linking and purifying input DNA phenol:chloroform: isoamyl alcohol-based precipitation method. Purified DNA was subjected to library preparation for sequencing or used for quantitative PCR.

## ChIP-qPCR

After ChIP and DNA extraction, an equal volume of purified DNA was used to perform qPCR. Primers used for the *Margin* promoter region were *Fwd* - 5'-AATAATGAAGTCGTGAAGAACAAA-3' and *Rev* - 5'-TTGTAACCGAGTAGAAGTTCAAT-3' and those for *SETD8* promoter region were *Fwd* - 5'-GACCGGGCTATTTCTTTTAAAAGAATATAAATAAA-CAAAGG-3' and *Rev* - 5'-CCTGTACAACTACTTAAAATGAAGACACGGA-3'. The primers are designed to span TCF/LEF-binding sites in the promoters of these two target genes. The desert regions have been described previously (Reddy et al, 2020). The TB Green Premix Ex Taq

II (Tli RNase H Plus) – (Takara Bio Inc.) was used to perform the qPCR on a ViiA 7 Real-Time PCR System (Applied Biosystems). The fold enrichment was calculated using the formula–Fold enrichment = $2^{(Ct\ (Target\ antibody)-Ct\ (IgG))}$.

## ChIP-seq library preparation and sequencing

A total of 1 ng of purified ChIPped DNA for each sample was used for library preparation using an Ultra II DNA kit (NEB) per the manufacturer's instructions. All library samples were amplified for 9–10 cycles depending on cycle number estimation by qPCR. Amplified libraries were subjected to double-sided bead purification using one round of 0.5X vol of Hi-prep PCR purification kit (Magbio Genomics) to remove the primer dimers and one round of 1.8X vol to remove the fragments larger than 1,000 bp. Library concentration was determined using Qubit (Thermo Fisher Scientific), and average fragment size was estimated using DNA HS assay on Bioanalyzer 2100 (Agilent Technologies) before pooling libraries at an equimolar ratio. 1.5 pM of the denatured libraries were used as an input to obtain sequencing reads using Nextseq 550 (Illumina) at IISER Pune.

## Sequencing and adaptor trimming

1.5 pM of the denatured libraries were used as an input to obtain sequencing reads. The DNA was loaded onto a NextSeq 500/550 High-Output v2.5 Kit (75 cycles) (Cat # 20024906; Illumina) and sequenced using 35 cycles for read1, 8 cycles for indexes, and 35 cycles for read2 on Nextseq 550 (Illumina) at the sequencing facility of IISER Pune.

After sequencing, the bcl files obtained were converted to fastq (Illumina).

bcl2fastq -R 210402_NB551653_0077_AHNCFTBGXF -p 10 –output-dir 210402_NB551653_0077_AHNCFTBGXF/fastq_files –sample-sheet 210402_NB551653_0077_AHNCFTBGXF/SampleSheet.csv

The sample-specific files were generated by concatenation of all four sequenced lanes.

cat 0-I_S11_L001_R1_001.fastq.gz 0-I_S11_L002_R1_001.fastq.gz 0-I_S11_L003_R1_001.fastq.gz 0-I_S11_L004_R1_001.fastq.gz > 0-I_R1.fastq.gz.

## ChIP-seq analysis

The quality of the fastq files was checked using FastQC (Andrews, 2010), and the quality control (QC) data were consolidated using MultiQC (Ewels et al, 2016) (Figs S32 and S33). The fastq files were aligned to the latest assembly of the *Hydra* genome using the bowtie2 aligner (Langmead & Salzberg, 2012). The SAM files were converted to BAM files using SAMtools (Li et al, 2009). The BAM files were sorted and indexed. The sorted BAM files were used to generate bigwig files using deeptools (Ramirez et al, 2016). The bam files for the H4K20me1 ChIP samples were normalized against the respective input bam file for each time point. To obtain the differential binding profile of H4K20me1 during regeneration and upon ALP treatment, MACS3 peak caller and DiffBind packages were used in Rstudio to obtain the peak profiles. The sorted BAM files from the sequencing experiments were indexed using SAMtools (Li et al, 2009). From the sorted BAM files, peaks were called for each

biological replicate using MACS3. Peaks for DMSO and ALP-treated polyps were obtained by normalizing against the ChIP-Input samples and for the 0, 3, 8, 12, and 24 hpa samples. From the prepared bam files, peak files, and sample datasheet, ChIPQC and DiffBind packages were used in Rstudio to call for differential peaks. Peaks were identified upon ALP treatment with respect to DMSO and at 3, 8, 12, and 24 hpa with respect to 0 hpa at a *P*-value threshold of 0.05 with DESeq2 and EdgeR. The differential analysis was performed against DMSO (for ALP treated set) and 0 hpa datasets (for regeneration time points) to obtain peaks of H4K20me1 involved in head regeneration and Wnt signaling. The combinatorial occupancy matrices were generated using the computeMatrix tool, and the plots were generated using the plotHeatmap tool in the deeptools package. A k-means clustering was also performed as part of plotting the occupancy matrices to cluster the set of genes having similar characteristic changes of H4K20me1 peaks across the locus. The output BED file from DeepTools, having the coordinates of genes in the four clusters, was used to analyze the chromatin accessibility, histone modifications, and transcriptional output of each cluster.

To understand the physiological significance of the genes with differential H4K20me1 occupancy near them, the transcript IDs were extracted, and the corresponding protein names from the latest version of the Hydra genome assembly were isolated. Upon obtaining the corresponding protein names for each of the transcripts around the differential H4K20me1 occupied peaks, a STRING network analysis for multiple protein molecular functions was done using version 11.5 (Szklarczyk et al, 2015).

## Data Availability

The ChIP-seq data discussed in this publication have been deposited in NCBI's Gene Expression Omnibus (Edgar et al, 2002; Barrett et al, 2013) and are accessible through GEO Series accession number GSE205918 (https://www.ncbi.nlm.nih.gov/geo/query/acc.cgi?acc=GSE205918).

## Supplementary Information

## Acknowledgements

We thank Galande Laboratory members, especially Ankita Sharma, for the useful discussions. We thank Ankita Sharma for her help with the sequencing runs. We thank Amarendranath Soory for the discussion and suggestions. The work was supported by intramural grants from the Indian Institute of Science Education and Research, Pune and Shiv Nadar University, Delhi-NCR, to S Galande. S Galande is also a recipient of the JC Bose Fellowship (JCB/2019/000013) by the Science and Engineering Research Board, Government of India. A Gungi was supported by a fellowship from the CSIR. S Saha is supported by a fellowship from CSIR, Government of India.

## Author Contributions

A Gungi: conceptualization, data curation, formal analysis, investigation, methodology, and writing—original draft, review, and editing.
S Saha: data curation, formal analysis, validation, investigation, visualization, methodology, and writing—review and editing.
M Pal: formal analysis, investigation, and methodology.
S Galande: conceptualization, formal analysis, supervision, funding acquisition, project administration, and writing—original draft, review, and editing.

## Conflict of Interest Statement

The authors declare that they have no conflict of interest.

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
