## [Reviewer comments · Life Science Alliance]

Life Science Alliance

H4K20me1 plays a dual role in transcriptional regulation of regeneration & axis patterning in Hydra

Akhila Gungi, Shagnik Saha, Mrinmoy Pal, and Sanjeev Galande

DOI: <https://doi.org/10.26508/lsa.202201619>

Corresponding author(s): Sanjeev Galande, Shiv Nadar University

Review Timeline:

Submission Date:	2022-07-20
Editorial Decision:	2022-08-30
Revision Received:	2022-12-11
Editorial Decision:	2023-01-19
Revision Received:	2023-02-19
Editorial Decision:	2023-02-20
Revision Received:	2023-02-27
Accepted:	2023-02-28

Scientific Editor: Novella Guidi

Transaction Report:

August 30, 2022

Re: Life Science Alliance manuscript #LSA-2022-01619-T

Prof. Sanjeev Galande
Indian Institute of Science Education and Research
Biology
Dr Homi Bhabha Road
Pashan
Pune, Maharashtra 411008
India

Dear Dr. Galande,

Thank you for submitting your manuscript entitled "H4K20me1 plays a dual role in transcriptional regulation of regeneration & axis patterning in Hydra" to Life Science Alliance. The manuscript was assessed by expert reviewers, whose comments are appended to this letter. We invite you to submit a revised manuscript addressing the Reviewer comments.

Thank you for this interesting contribution to Life Science Alliance. We are looking forward to receiving your revised manuscript.

Sincerely,

B. MANUSCRIPT ORGANIZATION AND FORMATTING:

Reviewer #1 (Comments to the Authors (Required)):

1. SETD8 is known to activate the wnt target genes by interacting with TCF during development of bilaterians, and this paper shows that this system is conserved even in non-bilaterian Hydra and that H4K20me1, which is regulated by SETD8, plays an important role in head generation. This paper also demonstrates the genome wide methylation state during wnt signaling activation and regeneration. Hydra is well-studied for their signaling pathways in regeneration and development, and an important model organism in evolutionary developmental biology as a sister group of bilaterians for discussing the general biological mechanisms. On the other hand, the role of methylation in development and regeneration has not yet been well studied. Thus this study can be said unique and pioneering, and has potential for publication in Life science alliance.

2. While the first half of this paper clearly shows that methylation by SETD8 is involved in head generation via wnt signaling through the careful experiments, the second half of the paper regarding the comprehensive analysis of methylation states shown in Fig4-6 is critically lacking in explanation needed. It is unclear whether the method is consistent with the purpose of this paper, which is to elucidate the involvement of methylation in head generation. There appears to be a discrepancy between what is shown in Fig4-6 and the main text in Results. In addition, it is also unclear on which results some of the sentences in Discussion are based. Therefore, in order for this paper to be accepted, these issues (as described below in detail) would need to be resolved.

p6 "H4K20me1 occupancy is context-dependent and is linked to specific signaling pathways"

Fig. 4C and Fig. 5C are not clear as to which genes are used in the analysis; for Fig. 4C, the genes whose methylation status is changed by ALP treatment compared to Cont, and for Fig. 5C, the genes whose methylation status is changed compared to Ohpa should be used in the analysis. Also, it is not clear what is meant by p7L19 "the Hippo signaling pathway had the least number of gene bodies with this modification".

L23~31 What the graphs in Figs 4D and 4E show and what is described in the text do not appear to match. It is also unclear which genes were used in this analysis. Given the purpose of this paper, genes with altered methylation status should be extracted.

p6L32-p7L8, Fig6: ATAC seq results

It is unclear which genes were used in the analysis; Materials and Methods does not describe the method either. In addition, the description of the level of transcription and chromatin accessibility for the genes in each cluster in the main text does not seem to be consistent with Fig6. For example, the authors say "The cluster 1 genes have the highest level of H4K20me1, the least intensity of the ATAC seq reads, and the lowest levels of transcription", but the peak height of transcription in Fig6 does not seem to be so different from other clusters. Also the authors say "the occupancy of H4K20me1 is also excluded from open chromatin regions displaying active transcription."(p10 L12), but it does not look that way in any of the clusters as seen in Fig6. The vertical axis in Fig6 (in particular, "Transcription") should be the same scale for comparison among CLSTRs.

3. The additional issues;

p5 L13 "the Wnt signaling pathway components were enriched (Fig. 2B)"

Does any statistical analysis support this result?

Fig 2B

The current presentation of DMSO and UNC at the same stage side by side makes it difficult to see the changes in DMSO and UNC over time. It would be easier to see both the difference between DMSO and UNC and their respective changes over time if DMSO and UNC were placed vertically.

p5 L15 "Brachyury has been established as a bonafide head specification marker. Brachyury in Hydra is a head organizer gene

involved in head morphogenesis."

Please cite appropriate references.

p5 L29 "The occurrence of both SETD8 and its target histone modification, H4K20me1, is relatively higher in the head and the foot of the polyps (Fig.2D)."

Fig. 2D shows that the head (around the oral pore) does indeed appear to be stained, but not the foot.

p6 L21 "We used this experimental paradigm of ectopic activation of the Wnt signalling pathway and checked the occupancy of H4K20me1 on the promoter regions of Margin and setd8. We observed an enhanced occupancy of the modification at the promoter regions of both the genes (Fig. 3F).

How it was verified should be described (ChIP-qPCR?). Also, more detail information such as the realtime PCR system, reagents, primers need to be described in the corresponding section of materials and methods (p. 16).

Discussion

As a whole, it is difficult to tell which results correspond to what is described in Discussion, so a Fig number should be added.

For examples;

p11 L6 "in the context of Hydra head regeneration, where there is a necessity for various signalling pathways to interact, the trend indicates that H4K20me1 is negatively correlated with the transcription of genes. "

p11 L16 "The specificity is at TF binding sites, and a deeper investigation of the TCF binding sites displayed distinct regulation of H4K20me1 at these motifs in response to Wnt signalling activation."

Fig 8 , p11 L15~

Which results support these discussions?

Materials and Methods

p15 L10

Please provide information of α -SETD8 antibody.

p15 "RNA sequencing"

Please provide information of sequencing, such as the sequencer, the number of cycles (read length) and the number of reads obtained.

p17-18

The authors listed all commands to run the analysis software, but need not be listed unless special options were set.

Reviewer #2 (Comments to the Authors (Required)):

Gungi et al. treat Hydra with a compound that is known to inhibit the H4K20me1 methyltransferase SETD8, and find that this impairs the regeneration process. They show that this correlates with the regulation of Wnt pathway genes, that SETD8 interacts with beta-catenin, and SETD8 is upregulated upon an increase in Wnt signaling. They then go on to profile H4K20me1, and show that it anti-correlates with chromatin accessibility at gene promoters.

If all points below are addressed, this manuscript describes a role of SETD8 in Hydra regeneration, as well as the genomic distribution of H4K20me1 across the Hydra genome, and its relationship with transcriptional regulation during regeneration.

Major points:

1. The authors use a compound which has been described as a chemical inhibitor of SETD8. It is not clear if this inhibitor is specific to SETD8 at the concentration used (200uM) in Hydra. They should perform experiments showing that they inhibit SETB8 specifically, or at least state clearly any off-target effects they observe. For example, in Figure EV2 they show a small reduction of H4K20me1 when treated with UNC0379 for 8h. Why is this specific to 8h? What concentration was used in this Figure? Did they test different concentrations? And what were the effects on other histone modifications? What about the effects on non-histone substrates, eg PCNA (<https://pubmed.ncbi.nlm.nih.gov/25032507/>)? Maybe they can try to inhibit SETD8 genetically, and compare the effects to the compound?
2. Figure 3F: The authors should show a control region where H4K20me1 does not increase. Also, since ChIP-seq was eventually performed, screen shots of these regions in the ChIP-seq data would be informative to show that the ChIP-qPCR and ChIP-seq data are comparable, as well as a more detailed analysis of H4K20me1 on Wnt (target) genes, and how this correlates

with gene expression (changes) on these genes. Especially since the authors claim that H4K20me1 can activate or repress genes, and overall H4K20me1 seems to correlate with reduced chromatin accessibility.

3. It seems to me that Figure 4 is really Figure 5, and Figure 4 is obsolete (except 4a). Indicate the y axis units on the metaplots (is it cpm?). How was the clustering performed? Are there any genes that are not in a cluster?

4. What is the exact analysis underlying Figure 4B? As I understand it simply shows the number of genes associated with a particular pathway. The authors state that the Hippo pathway is underrepresented in H4K20me1 covered genes, but it also seems to have less genes in cluster 4, which as far as I understand is all genes that are not H4K20me1 covered? The authors should perform a statistical analysis to check for enrichments of genes with certain GO terms in their clusters.

5. Figure 6: While I can see an increase in ATAC seq signal, I cannot see increased transcription in cluster 3 and 4 vs cluster 1 and 2 genes.

6. It would be good to perform ChIP for total H3 or H4 and compare the profiles (heatmaps) to the h4K20me1 profiles, to exclude that the H4K20me1 profiles are specific for the modification and do not simply reflect histone binding.

Minor points:

1. The authors should provide more details on the analysis of the RNAseq experiments: PCA plots of all samples, how was the differential expression calculated, volcano plots, a list of all genes with their logFCs and adjusted p-values,...If regeneration is delayed by inhibitor treatment, then how can the authors be sure they are not just comparing different points in the regeneration process?

2. Figure 2D: The staining at the foot regions are very difficult to see.

3. In general, the y-axis scales of metaplots should be kept the same across the plots of a particular data type to make it easier to compare levels. The y-axis units should be indicated.

4. The authors could further explore the H4K20me1 distribution in the genome beyond gene bodies, for example at enhancers, or repetitive elements.

Reviewer #1:

1. SETD8 is known to activate the wnt target genes by interacting with TCF during development of bilaterians, and this paper shows that this system is conserved even in non-bilaterian Hydra and that H4K20me1, which is regulated by SETD8, plays an important role in head generation. This paper also demonstrates the genome wide methylation state during wnt signaling activation and regeneration. Hydra is well-studied for their signaling pathways in regeneration and development, and an important model organism in evolutionary developmental biology as a sister group of bilaterians for discussing the general biological mechanisms. On the other hand, the role of methylation in development and regeneration has not yet been well studied. Thus this study can be said unique and pioneering, and has potential for publication in Life science alliance.

2. While the first half of this paper clearly shows that methylation by SETD8 is involved in head generation via wnt signaling through the careful experiments, the second half of the paper regarding the comprehensive analysis of methylation states shown in Fig4-6 is critically lacking in explanation needed. It is unclear whether the method is consistent with the purpose of this paper, which is to elucidate the involvement of methylation in head generation. There appears to be a discrepancy between what is shown in Fig4-6 and the main text in Results. In addition, it is also unclear on which results some of the sentences in Discussion are based. Therefore, in order for this paper to be accepted, these issues (as described below in detail) would need to be resolved.

Response: We thank the reviewer for encouraging comments and suggestions on our manuscript. The further analysis suggested has helped in refining the manuscript and shaping the discussion around the role of H4K20me1 in a better manner.

p6 "H4K20me1 occupancy is context-dependent and is linked to specific signaling pathways"

Fig. 4C and Fig. 5C are not clear as to which genes are used in the analysis; for Fig. 4C, the genes whose methylation status is changed by ALP treatment compared to Cont, and for Fig. 5C, the genes whose methylation status is changed compared to 0hpa should be used in the analysis. Also, it is not clear what is meant by p7L19 "the Hippo signaling pathway had the least number of gene bodies with this modification".

Response: Thank you for this suggestion. We have performed the analysis of the ChIP seq data sets, retrieving the regions with differential occupancy of H4K20me1 in both ALP treatment and regeneration time points (Fig. 4-6). We have also done different analyses to understand the significance of the changing histone mark occupancy on the genomic regions in relation to the genes associated with it and this is included in the revised version of the manuscript (Fig. 4-6).

L23~31 What the graphs in Figs 4D and 4E show and what is described in the text do not appear to match. It is also unclear which genes were used in this analysis. Given the purpose of this paper, genes with altered methylation status should be

extracted.

Response: As suggested by this and the second reviewer, we have retrieved the genes with altered methylation status and interpreted the results in this condition. Following this, we retrieved the genes and performed further annotation of genes, GO term enrichment, and STRING interaction analysis (Lines 207-216). There was no significant GO term enrichment in the ALP condition and we have therefore depicted the STRING results in the revised manuscript (Fig. 4C-F; Fig. S9-S24).

p6L32-p7L8, Fig6: ATAC seq results

It is unclear which genes were used in the analysis; Materials and Methods does not describe the method either. In addition, the description of the level of transcription and chromatin accessibility for the genes in each cluster in the main text does not seem to be consistent with Fig6. For example, the authors say "The cluster 1 genes have the highest level of H4K20me1, the least intensity of the ATAC seq reads, and the lowest levels of transcription" , but the peak height of transcription in Fig6 does not seem to be so different from other clusters. Also the authors say "the occupancy of H4K20me1 is also excluded from open chromatin regions displaying active transcription."(p10 L12), but it does not look that way in any of the clusters as seen in Fig6. The vertical axis in Fig6 (in particular, "Transcription") should be the same scale for comparison among CLSTRs.

Response: We thank the reviewer for this comment and suggestion. We have performed this analysis too on the differentially methylated regions of the *Hydra* genome and associated genes. We have maintained the scale of the vertical axis constant for each type of data we have analysed to enable comparison between the clusters (Fig. 4A-B; Fig. 5; Fig. 6A-K). The results in the revised manuscript now denote the differentially methylated genomic regions.

3. The additional issues;

p5 L13 "the Wnt signaling pathway components were enriched (Fig. 2B)"

Does any statistical analysis support this result?

Response: The enrichment is based on the proportion of the genes with H4K20me1 occupancy which is statistically supported. However, in the revised version of the manuscript, we have retrieved the differentially occupied regions and performed an interaction analysis on the associated genes (Fig. 4C-F; Fig S9-S24).

Fig 2B

The current presentation of DMSO and UNC at the same stage side by side makes it difficult to see the changes in DMSO and UNC over time. It would be easier to see both the difference between DMSO and UNC and their respective changes over time if DMSO and UNC were placed vertically.

Response: We have modified the representation of the figure so that the time course of regeneration is displayed in the two different conditions – DMSO and UNC

treated. We agree with the reviewer that this makes the visualisation of the molecular dynamics easy and thank the reviewer for this suggestion (Fig. 2B).

p5 L15 "Brachyury has been established as a bonafide head specification marker. Brachyury in Hydra is a head organizer gene involved in head morphogenesis."

Please cite appropriate references.

Response: Thank you for pointing this out; the appropriate references have been added (Lines 114-115).

p5 L29 "The occurrence of both SETD8 and its target histone modification, H4K20me1, is relatively higher in the head and the foot of the polyps (Fig.2D)."

Fig. 2D shows that the head (around the oral pore) does indeed appear to be stained, but not the foot.

Response: We have now used images of polyps with a clearer representation of the increased occurrence of the KMT and its histone mark at both the poles (Fig. 2D). We have also provided more images in the supplementary data to support this (Fig. S5).

p6 L21 "We used this experimental paradigm of ectopic activation of the Wnt signalling pathway and checked the occupancy of H4K20me1 on the promoter regions of Margin and setd8. We observed an enhanced occupancy of the modification at the promoter regions of both the genes (Fig. 3F).

How it was verified should be described (ChIP-qPCR?). Also, more detail information such as the realtime PCR system, reagents, primers need to be described in the corresponding section of materials and methods (p. 16).

Response: We have provided the relevant information in the Methods section of the revised manuscript (Lines 472-482).

Discussion

As a whole, it is difficult to tell which results correspond to what is described in Discussion, so a Fig number should be added.

For examples;

p11 L6 "in the context of Hydra head regeneration, where there is a necessity for various signalling pathways to interact, the trend indicates that H4K20me1 is negatively correlated with the transcription of genes. "

p11 L16 "The specificity is at TF binding sites, and a deeper investigation of the TCF binding sites displayed distinct regulation of H4K20me1 at these motifs in response to Wnt signalling activation."

Fig 8, p11 L15~

Which results support these discussions?

Response: This suggestion has indeed helped increase clarity the manuscript. As asked above:

1. "in the context of Hydra head regeneration, where there is a necessity for various signalling pathways to interact, the trend indicates that H4K20me1 is negatively correlated with the transcription of genes. " (in the revised manuscript Fig. 6A, 6B indicating an increase of transcription upon inhibition of SETD8 by UNC0379, Fig. S9-S24 indicating the large network of genes with differential occupancy of H4K20me1 vs the STRING interaction networks in Fig. 4C-4E showing very few genes with differential methylation).
2. "The specificity is at TF binding sites, and a deeper investigation of the TCF binding sites displayed distinct regulation of H4K20me1 at these motifs in response to Wnt signalling activation." (in the revised manuscript Fig. 3F shows occupancy at localised TCF motifs studied using ChIP-qPCRs)
3. The interpretations of Fig 7B, and Fig. 7C (Fig.8) in the revised manuscript have been obtained based on the ChIP seq analysis done in Fig4-Fig.6 comparing the two physiological conditions of Wnt activation and regeneration.

In the revised manuscript, we have clearly cited all the results at the places where they are being discussed (Figure citations in Discussion section).

Materials and Methods

p15 L10

Please provide information of α -SETD8 antibody.

p15 "RNA sequencing"

Please provide information of sequencing, such as the sequencer, the number of cycles (read length) and the number of reads obtained.

p17-18

The authors listed all commands to run the analysis software, but need not be listed unless special options were set.

Response: We have provided all the relevant information about the antibody, sequencer, and sequencing reads in the methods section of the revised manuscript (Lines 424-425, 441-445). The codes used did not require custom options and, as suggested, have been removed to avoid redundant information.

Reviewer #2:

Gungi et al. treat Hydra with a compound that is known to inhibit the H4K20me1 methyltransferase SETD8, and find that this impairs the regeneration process. They show that this correlates with the regulation of Wnt pathway genes, that SETD8 interacts with beta-catenin, and SETD8 is upregulated upon an increase in Wnt signaling. They then go on to profile H4K20me1, and show that it anti-correlates with chromatin accessibility at gene promoters.

If all points below are addressed, this manuscript describes a role of SETD8 in Hydra regeneration, as well as the genomic distribution of H4K20me1 across the Hydra genome, and its relationship with transcriptional regulation during regeneration.

Major points:

1. The authors use a compound which has been described as a chemical inhibitor of SETD8. It is not clear if this inhibitor is specific to SETD8 at the concentration used (200uM) in Hydra. They should perform experiments showing that they inhibit SETD8 specifically, or at least state clearly any off-target effects they observe. For example, in Figure EV2 they show a small reduction of H4K20me1 when treated with UNC0379 for 8h. Why is this specific to 8h? What concentration was used in this Figure? Did they test different concentrations? And what were the effects on other histone modifications? What about the effects on non-histone substrates, eg PCNA (<https://pubmed.ncbi.nlm.nih.gov/25032507/>)? Maybe they can try to inhibit SETD8 genetically, and compare the effects to the compound?

Response: Thank you for this insightful question. We have performed time-course experiments and observed the effects of various concentrations of the compound UNC0379. At a higher concentration of 250 μ M, the polyps start to disintegrate within 4 h and cannot be used for performing regeneration assays. At the concentration used for regeneration assays (150 μ M), the polyps were harvested at different time points to establish the duration of treatment for regeneration assays. We have mentioned this information in the revised version of the manuscript. Further, we are aware of the compounding effects on the function of SETD8 on its non-histone substrates, but we still attribute this to effects on catalytic function. Such effects will also be present upon genetic inhibition and more extensive assays will be required to delineate these that are out of the scope of this manuscript due to time constraints. We have tried genetically inhibiting the function using siRNA-mediated knockdowns. However, unlike transcription factor knockdowns that are quite stable (Reddy et. al. 2019), knockdown of epigenetic regulators, and specifically histone methyltransferases, is very transient, and the feedback regulation does not allow enough loss of the enzyme expression to affect morphogenetic events. Regeneration assays performed after knockdown suggest the same, where the effect on regeneration is less pronounced than with chemical inhibition. The data is shown below.

2. Figure 3F: The authors should show a control region where H4K20me1 does not increase. Also, since ChIP-seq was eventually performed, screen shots of these regions in the ChIP-seq data would be informative to show that the ChIP-qPCR and ChIP-seq data are comparable, as well as a more detailed analysis of H4K20me1 on Wnt (target) genes, and how this correlates with gene expression (changes) on these genes. Especially since the authors claim that H4K20me1 can activate or repress genes, and overall H4K20me1 seems to correlate with reduced chromatin accessibility.

Response: We thank the reviewer for this comment. We have shown a genomic desert region (used in Reddy et. al. 2020) as a control that does not display altered H4K20me1 status. Also, in the analysis we have performed for the revised version of the manuscript, we have retrieved and plotted the IGV screenshots for the genomic regions in different clusters (Fig. 4C-F, Fig. 6L-O). Isolating Wnt-related genes with high accuracy is difficult due to the lack of complete annotation of the genome, and we have tried to do this using the new version of the *Hydra* genome available (v 3). However, in the ALP-treated polyps, we do find a positive correlation of H4K20me1 occupancy with few genomic regions around Wnt target transcription factors (Fig. 4). The negative correlation with transcription is more prominent in the regeneration phenotype, where upon inhibition of SETD8, we observed increased transcription at multiple time points (Fig. 5 and Fig. 6).

3. It seems to me that Figure 4 is really Figure 5, and Figure 4 is obsolete (except 4a). Indicate the y axis units on the metaplots (is it cpm?). How was the clustering performed? Are there any genes that are not in a cluster?

Response: Figure 4 depicts the dynamics of H4K20me1 in ALP-treated polyps, while Figure 5 shows this in regenerating tips of the polyps. In the revised version of the manuscript, we have only shown the differentially occupied regions in the main figures and the whole genome occupancy in the extended information (Fig. S7 and Fig. S8). The clustering was performed using Deeptools and it was a k-means based clustering showing the occupancy across treatment conditions or timepoints. Since the previous analysis did not look at differentially occupied regions, almost all the genes were displayed in the heatmap. The detailed analysis we have performed to identify the differentially occupied genomic regions has allowed us to differentiate between the behaviour of H4K20me1 in both physiological conditions and make better interpretations. This has now been incorporated in the revised manuscript. (Lines 163-216; Lines 270-288).

4. What is the exact analysis underlying Figure 4B? As I understand it simply shows the number of genes associated with a particular pathway. The authors state that the Hippo pathway is underrepresented in H4K20me1 covered genes, but it also seems to have less genes in cluster 4, which as far as I understand is all genes that are not H4K20me1 covered? The authors should perform a statistical analysis to check for enrichments of genes with certain GO terms in their clusters.

Response: In the original version, we retrieved the number of genes associated with specific GO terms with occupancy of H4K20me1. However, for more physiological relevance to the process being investigated, we have performed a differential occupancy analysis and retrieved the genomic regions and genes associated with them. Since we do not find significant GO term enrichment due to the lack of good annotation of the genes in *Hydra*, we performed a STRING interaction analysis to identify significantly interacting networks with differentially H4K20 mono-methylated regions (Fig. 4C-F; Fig. S9-S24).

5. Figure 6: While I can see an increase in ATAC seq signal, I cannot see increased transcription in cluster 3 and 4 vs cluster 1 and 2 genes.

Response: We have now retrieved the differentially occupied regions with respect to the 0 h time point during regeneration and performed further analysis on these regions (Fig. 5A-C; Fig. 6A-D). To understand the effect of H4K20me1 on transcription, we have also plotted the UNC-treated RNA seq data (Fig. 6E-H). This has allowed us to better understand the role of H4K20me1 in transcriptional regulation. The discussion has been added in the revised version of the manuscript.

6. It would be good to perform ChIP for total H3 or H4 and compare the profiles (heatmaps) to the H4K20me1 profiles, to exclude that the H4K20me1 profiles are specific for the modification and do not simply reflect histone binding.

Response: The reviewer has raised an important point here. Our previously published report on understanding enhancer-mediated regulation of Wnt signalling (Reddy & Gungl et. al., *Epigenetics & Chromatin* 2020) includes the data for H3 ChIPseq and demonstrates a conserved loss of H3 occupancy at the +1 nucleosome in *Hydra*. Comparing both the profile and the level of H4K20me1 occupancy to that, we are able to ensure that we are not observing general histone binding. H4K20me1 is a very low occurrence histone mark and does not display the +1 nucleosome loss as seen with H3. Further, we see a clear spreading of H4K20me1 occupancy along the gene bodies with differential occupancy at exons and introns as well. This needs deeper analysis to decipher the role of H4K20me1 in splicing, gene expression regulation, or isoform regulation, and will be a worthwhile new line of investigation. We therefore believe this is currently out of scope for this manuscript.

Minor points:

1. The authors should provide more details on the analysis of the RNAseq experiments: PCA plots of all samples, how was the differential expression calculated, volcano plots, a list of all genes with their logFCs and adjusted p -

values,...If regeneration is delayed by inhibitor treatment, then how can the authors be sure they are not just comparing different points in the regeneration process?

Response: Thank you for pointing this out. We have now provided the relevant information in the extended information of the revised manuscript. From the volcano plots obtained comparing the same time points in control and treated conditions, we can clearly see changes in gene expression that are due to the inhibition of SETD8 (Fig. S25). The plots in the main figure panel represent the logCPM+1 values of genes that show differential expression to depict the dynamics.

2. Figure 2D: The staining at the foot regions are very difficult to see.

Response: We thank the reviewer for pointing this out. We have changed the representative images to reflect better the occurrence of both SETD8 and the histone marks in the head and foot of the polyps (Fig. 2D, Fig. S5). We have also provided additional images in the extended information as evidence for the same.

3. In general, the y-axis scales of metaplots should be kept the same across the plots of a particular data type to make it easier to compare levels. The y-axis units should be indicated.

Response: We thank the reviewer for this comment. As suggested by you and the first reviewer, we have kept the y-axis of all the plots the same for a single type of data to enable comparison across clusters obtained in the new analysis. These revised figures are provided in the revised version of the manuscript (Fig. 4,5,6).

4. The authors could further explore the H4K20me1 distribution in the genome beyond gene bodies, for example at enhancers, or repetitive elements.

Response: We thank the reviewer for this insightful suggestion. We have retrieved the intergenic, intronic, and promoter sequences for the whole genome of *Hydra* using the available version of the genome and Bedtools. However, we did not find significantly differential trends of H4K20me1 occupancy in these regions and have therefore not pursued this further. However, further analysis could provide us with information to understand the role of H4K20me1 in regulating enhancer regions, and we plan to explore this in the future.

January 19, 2023

Re: Life Science Alliance manuscript #LSA-2022-01619-TR

Prof. Sanjeev Galande
Indian Institute of Science Education and Research Pune
Biology
Dr Homi Bhabha Road
Pashan
Pune, Maharashtra 411008
India

Dear Dr. Galande,

Thank you for submitting your revised manuscript entitled "H4K20me1 plays a dual role in transcriptional regulation of regeneration & axis patterning in Hydra" to Life Science Alliance. The manuscript has been seen by the original reviewers whose comments are appended below. While the reviewers continue to be overall positive about the work in terms of its suitability for Life Science Alliance, some important issues remain.

Our general policy is that papers are considered through only one revision cycle; however, given that the suggested changes are relatively minor, we are open to one additional short round of revision. Please note that I will expect to make a final decision without additional reviewer input upon resubmission.

Please submit the final revision within one month, along with a letter that includes a point by point response to the remaining reviewer comments.

To upload the revised version of your manuscript, please log in to your account: <https://lsa.msubmit.net/cgi-bin/main.plex>
You will be guided to complete the submission of your revised manuscript and to fill in all necessary information.

B. MANUSCRIPT ORGANIZATION AND FORMATTING:

Sincerely,

Reviewer #1 (Comments to the Authors (Required)):

1.

In "H4K20me1 occupancy is context-dependent and is linked to specific signalling pathways" (Figs. 4-6), the methylation state changes during wnt activation and regeneration are more clearly defined by the new analysis of the differentially H4K20me1 occupied regions. The other points have been appropriately corrected according to the suggestions.

2.

I have the following two questions/suggestions regarding the new analysis.

(1) L205 Interestingly, across all timepoints, we observed a decrease in the occupancy of the activation-associated histone mark with respect to the 0 h time point of regeneration (Fig. 5D-L).

Was the occupancy of activation-associated histone marks significantly reduced only in the regions where H4K20me1 occupancy altered? And does this decrease correlate with an increase or decrease in transcription? These are important information to understand if there is a relationship between H4K20me1 and H4K20me2/me3 and if there is regulation of transcription by H4K20me2/me3 discussed in Discussion (L348)

(2) The authors showed the genes with increased H4K20me1 occupancy by ALP treatment showed increased transcription (Fig. 4), while inhibition of SETD8 by UNC treatment increased transcription of these genes (Fig. 6). I understand that H4K20me1 plays a dual role in transcriptional regulation, but does it depend on the type of gene or the situation (such as cellular processes) as to whether transcription is positively or negatively regulated by H4K20me1? At least for wnt-related genes which showed an increase in H4K20me1 occupancy (Fig1 and 4), it should be noted what it was like during regeneration (Fig5, 6).

3. Minor points:

(1) Some Fig numbers in Results are not correct. Please check. (eg. Figure A~F)

(2) Please describe the K-means clustering and STRING interaction analysis in Materials and Methods.

Reviewer #2 (Comments to the Authors (Required)):

In general the manuscript has been improved and some major points clarified, as well as all minor points addressed. However, the genome wide effects of Setd8 inhibition on transcription are still a bit unclear given the Figures presented. Maybe the authors could improve the transcription analysis or, if the effects remain unclear, explain this in the discussion. Below my responses to individual major points.

1. The authors show that siRNA knock-downs are not sufficient to cause the phenotype. It would be appropriate to show these data in the Supplement and mention the caveats of using a compound to inhibit a protein function rather than a genetic mechanism in the discussion.

2. I was more thinking about screen shots for the same regions that are targeted in the qPCR. Nevertheless, I appreciate the new Figure 4 and the clarifications it brings. I would suggest to plot the RNAseq results in a different way, for example as boxplots of cpm per gene, as the exon-intron distribution per gene is different and therefore makes the plots aligned at TSS/TSS look a bit confusing.

3. Thank you for the clarifications.

4. Thank you for providing the STRING analysis. I am not sure how to interpret these results, and if there are any significant differences between different groups.

5. I still do not see large changes in RNAseq read distributions. However, plotting the RNAseq data differently might help.

6. Thank you for the clarifications.

Reviewer #1 (Comments to the Authors (Required)):

1. In "H4K20me1 occupancy is context-dependent and is linked to specific signalling pathways" (Figs. 4-6), the methylation state changes during wnt activation and regeneration are more clearly defined by the new analysis of the differentially H4K20me1 occupied regions. The other points have been appropriately corrected according to the suggestions.

Response: We thank the reviewer for the suggestion to perform the differential occupancy analysis and for further queries that improved the manuscript.

2. I have the following two questions/suggestions regarding the new analysis.

(1) L205 Interestingly, across all timepoints, we observed a decrease in the occupancy of the activation-associated histone mark with respect to the 0 h time point of regeneration (Fig. 5D-L).

Was the occupancy of activation-associated histone marks significantly reduced only in the regions where H4K20me1 occupancy altered? And does this decrease correlate with an increase or decrease in transcription? These are important information to understand if there is a relationship between H4K20me1 and H4K20me2/me3 and if there is regulation of transcription by H4K20me2/me3 discussed in Discussion (L348).

Response: The occupancy of the activation associated histone marks is altered on many regions in the genome upon activation of Wnt signalling. We have shown this in our previous work elaborating the enhancer-mediated Wnt gene activation (Reddy et al, 2020). Here, we focused on regions with differential H4K20me1 occupancy to understand the role of this relatively less-studied histone mark in *Hydra* physiology.

(2) The authors showed the genes with increased H4K20me1 occupancy by ALP treatment showed increased transcription (Fig. 4), while inhibition of SETD8 by UNC treatment increased transcription of these genes (Fig. 6). I understand that H4K20me1 plays a dual role in transcriptional regulation, but does it depend on the type of gene or the situation (such as cellular processes) as to whether transcription is positively or negatively regulated by H4K20me1? At least for wnt-related genes which showed an increase in H4K20me1 occupancy (Fig1 and 4), it should be noted what it was like during regeneration (Fig5, 6).

Response: We thank the reviewer for this query. We investigated the occupancy of H4K20me1 during regeneration on the genomic regions with increased occupancy upon ALP treatment. This corresponds to cluster 3 of the differentially occupied regions due to ALP treatment. We do not observe significant differences in the occupancy of the histone modification during the regeneration time course. Therefore, we believe that during ALP treatment as the Wnt pathway is being ectopically activated throughout the polyp, other signalling pathways presumably do

not play any role towards this phenotype. However, during regeneration, there is a larger dynamicity in the signalling networks being reprogrammed and H4K20me1 plays a different role in different cell types based on the context. We have included this data in the supplementary information and this adds to the discussion already included in the manuscript (Supplementary Figure: S30; Discussion: Lines 334-338).

3. Minor points:

(1) Some Fig numbers in Results are not correct. Please check. (eg. Figure A~F)

Response: We have checked all figure citations and corrected as necessary.

(2) Please describe the K-means clustering and STRING interaction analysis in Materials and Methods.

Response: We have now added the description of the methods for k-means and STRING interaction analysis in the Materials and Methods (Lines: 559-569).

Reviewer #2 (Comments to the Authors (Required)):

In general the manuscript has been improved and some major points clarified, as well as all minor points addressed. However, the genome wide effects of Setd8 inhibition on transcription are still a bit unclear given the Figures presented. Maybe the authors could improve the transcription analysis or, if the effects remain unclear, explain this in the discussion. Below my responses to individual major points.

Response: We thank the reviewer for the insightful suggestions.

1. The authors show that siRNA knock-downs are not sufficient to cause the phenotype. It would be appropriate to show these data in the Supplement and mention the caveats of using a compound to inhibit a protein function rather than a genetic mechanism in the discussion.

Response: We thank the reviewer for suggesting incorporation of the data of siRNA-mediated knockdown in the manuscript. We have added the same in the Supplementary information (Fig. S31). We have also detailed the caveats of using a small molecule inhibitor instead of a genetic mechanism in the discussion (Discussion: Lines 249-258).

2. I was more thinking about screen shots for the same regions that are targeted in the qPCR. Nevertheless, I appreciate the new Figure 4 and the clarifications it brings. I would suggest to plot the RNAseq results in a different way, for example as boxplots of cpm per gene, as the exon-intron distribution per gene is different and therefore makes the plots aligned at TSS/TSS look a bit confusing.

Response: We thank the reviewer for suggesting a different way of representing the RNA-seq results. This has enabled us to discuss regarding the transcription of specific gene sets when Wnt signalling is ectopically activated and during regeneration. We have incorporated this data in the supplementary and retained the

global analysis in the main figures. We have discussed this data in the revised manuscript (Results: Lines 190-194; 222-230; Supplementary Figures S9, S10, S28, S29).

3. Thank you for the clarifications.

4. Thank you for providing the STRING analysis. I am not sure how to interpret these results, and if there are any significant differences between different groups.

Response: We performed the STRING analysis to understand if any signalling pathways emerge from the gene lists that we obtained after identifying the differentially H4K20me1 methylated regions of the genome. We observe genes functioning in various pathways emerging in this analysis. These genes significantly associate with the differentially modified genomic regions and, therefore may be associated with the different cellular processes like proliferation, differentiation, or apoptosis and the underlying metabolism of the cells during these processes.

5. I still do not see large changes in RNAseq read distributions. However, plotting the RNAseq data differently might help.

Response: We have plotted the RNA-seq results for regenerating polyps with and without inhibition of SETD8 using the alternate method suggested by the reviewer (Fig. S9, Fig. S10, Fig. S28, Fig. S29). The changes in the read counts are now visible and indicate the differential transcription of the genes important for physiological processes. We have also added the interpretations of these results in the revised manuscript (Results: Lines 190-194; 221-230).

6. Thank you for the clarifications.

February 20, 2023

RE: Life Science Alliance Manuscript #LSA-2022-01619-TRR

Prof. Sanjeev Galande
Indian Institute of Science Education and Research Pune
Biology
Dr Homi Bhabha Road
Pashan
Pune, Maharashtra 411008
India

Dear Dr. Galande,

Thank you for submitting your revised manuscript entitled "H4K20me1 plays a dual role in transcriptional regulation of regeneration & axis patterning in Hydra". We would be happy to publish your paper in Life Science Alliance pending final revisions necessary to meet our formatting guidelines.

- please upload your supplementary figures as single files
- please add a conflict of interest statement to the main manuscript text
- please add the supplementary figure legends to the main manuscript text
- please double-check your figure callouts in your manuscript text; you have callouts for Figures A, Figure E, etc., but the figure number is not designated
- please add figure callouts for Fig 1 A, B; Fig 3 A, B; Fig 6 I, J, K; Fig 7 A to your main manuscript text

Figure Check:

- Fig S1: please describe each panel in the figure legend
- please add/make more visible scale bars throughout for images like Figure 1C and Figure 2D

A. FINAL FILES:

B. MANUSCRIPT ORGANIZATION AND FORMATTING:

Sincerely,

February 28, 2023

RE: Life Science Alliance Manuscript #LSA-2022-01619-TRRR

Prof. Sanjeev Galande
Shiv Nadar University
Life Sciences
NH-91, Gautam Buddha Nagar
Greater Noida, Uttar Pradesh 201314
India

Dear Dr. Galande,

Thank you for submitting your Research Article entitled "H4K20me1 plays a dual role in transcriptional regulation of regeneration & axis patterning in Hydra". It is a pleasure to let you know that your manuscript is now accepted for publication in Life Science Alliance. Congratulations on this interesting work.

DISTRIBUTION OF MATERIALS:

Again, congratulations on a very nice paper. I hope you found the review process to be constructive and are pleased with how the manuscript was handled editorially. We look forward to future exciting submissions from your lab.

Sincerely,
